



# Stable water isotope signals in tropical ice clouds in the West African monsoon simulated with a regional convection-permitting model

Andries Jan de Vries[1], Franziska Aemisegger[1], Stephan Pfahl[2], Heini Wernli[1]

[1]ETH Zürich, Institute for Atmospheric and Climate Science, Zürich, Switzerland
[2]Institute of Meteorology, Freie Universität Berlin, Berlin, Germany

*Correspondence to*: Andries Jan de Vries (andries.devries@env.ethz.ch)

**Abstract.** Tropical ice clouds have an important influence on the Earth's radiative balance. They often form as a result of tropical deep convection, which strongly affects the water budget of the tropical tropopause layer. Ice cloud formation involves complex interactions on various scales, which are not fully understood yet and lead to large uncertainties in climate predictions.

In this study, we investigate the formation of tropical ice clouds related to deep convection in the West African monsoon, using stable water isotopes as tracers of moist atmospheric processes. We perform simulations using the regional isotope-enabled model COSMO$_{iso}$ with different resolutions and treatments of convection for the period of June-July 2016. First, we evaluate the ability of our simulations to represent the isotopic composition of monthly precipitation through comparison with GNIP observations, and the precipitation characteristics related to the monsoon evolution and convective storms based on insights

from the DACCIWA field campaign in 2016. Next, a case study of a mesoscale convective system (MCS) explores the isotope signatures of tropical deep convection in atmospheric water vapour and ice. Convective updrafts within the MCS inject enriched ice into the upper troposphere leading to depletion of vapour within these updrafts due to the preferential condensation and deposition of heavy isotopes. Water vapour in downdrafts within the same MCS are enriched by non-fractionating sublimation of ice. In contrast to ice within the MCS core regions, ice in widespread cirrus shields is isotopically in approximate

equilibrium with the ambient vapour, which is consistent with in situ formation of ice. These findings from the case study are supported by a statistical evaluation of isotope signals in the West African monsoon ice clouds. The following five key processes related to tropical ice clouds can be distinguished based on their characteristic isotope signatures: (1) convective lofting of enriched ice into the upper troposphere, (2) cirrus clouds that form in situ from ambient vapour under equilibrium fractionation, (3) sedimentation and sublimation of ice in the mixed-phase cloud layer in the vicinity of convective systems

and underneath cirrus shields, (4) sublimation of ice in convective downdrafts that enriches the environmental vapour, and (5) the freezing of liquid water in the mixed-phase cloud layer at the base of convective updrafts. Importantly, the results show that convective systems strongly modulate the humidity budget and the isotopic composition of the lower tropical tropopause layer. They contribute to about 40% of the total water and 60% of HDO in the 175-125 hPa layer in the African monsoon region according to estimates based on our model simulations. Overall, this study demonstrates that isotopes can serve as

useful tracers to disentangle the role of different processes in the Earth's water cycle, including convective transport, the formation of ice clouds, and their impact on the tropical tropopause layer.



## 1 Introduction

Tropical ice clouds are a key element of the climate system. They have an important influence on the Earth's radiative budget through reflecting shortwave radiation back to space and trapping longwave radiation underneath (Ackerman et al., 1988). A

key process that leads to the formation of tropical ice clouds is tropical convection (Massie et al., 2002; Luo and Rossow, 2004). Tropical convective systems and cloud ice that detrains from these systems strongly influences the humidity budget of the cold tropopause region; the tropical tropopause layer (TTL; Fueglistaler et al., 2009). A complex interplay of different processes controls the water transport through this region into the lower stratosphere, which is a theme of major scientific interest as stratospheric water has a large impact on the climate system through its influence on the radiative balance and

stratospheric chemistry (Notholt et al., 2010; Randel and Jensen, 2013; Dessler et al., 2016).

One of the Earth's region where convective systems are particularly strong and can affect the TTL, is West Africa (Zipser et al., 2006; Cetrone and Houze, 2009; Fierli et al., 2011). In boreal summer, the atmospheric circulation over this region is dominated by the West African monsoon (WAM). The WAM system (Fig. 1) involves a complex interplay between different

large-scale circulation and mesoscale convective processes (Redelsperger et al., 2002; Lacour et al., 2017; Diekmann et al., 2021). Intense solar heating over the Saharan desert leads to the formation of the Saharan heat low (Lavaysse et al., 2009), while much colder sea surface temperatures in the eastern tropical Atlantic cool the air near the surface. The resulting pressure gradient drives an inflow of moist air from the ocean into the continent that feeds rainfall over this region. The tropical rain belt stretches zonally over Africa and is along its northern edge confined by the mid tropospheric anticyclone aloft the Saharan

heat low. The WAM follows a distinct seasonal evolution (Thorncroft et al., 2011). Before its onset, the rain belt is anchored over the coast of Guinea, and migrates abruptly northward into the Sahel after its onset that usually takes place in the period of late June to early July (Sultan and Janicot, 2003; FitzPatrick et al., 2015). Precipitation over West Africa can stem from different types of convection embedded in the WAM circulation, whereby highly organized mesoscale convective systems (MCSs) contribute up to 90% of rainfall (Mathon et al., 2002). MCSs typically traverse over North Africa in a westward

direction, often along instabilities in the African easterly jet, so-called African easterly waves (Fink and Reiner, 2003; Knippertz et al., 2017).

MCSs over West Africa are not only of central importance for precipitation, but also for the formation of tropical ice clouds (Mathon and Laurent, 2001; Frey et al., 2011). Large MCSs are the most important contributor of tropical ice clouds over this

region in summer (Yuan and Houze, 2010). Convective systems inject ice into the upper troposphere, which is detrained into large anvils that spread out into thinning cirrus shields (Luo and Rossow, 2004). Other types of ice clouds form directly from the vapour phase due to the uplift from gravity waves or above convective systems (Pfister et al., 2001). These two formation pathways of ice clouds were later coined as liquid-origin and in situ cirrus (Krämer et al., 2016, 2020; Luebke et al., 2016; Wernli et al., 2016; Gasparini et al., 2018). Liquid-origin refers to the freezing of liquid water in updrafts within the mixed-



phase cloud layer, while in situ refers to the formation of ice crystals directly from the vapour phase. Other classifications of
cirrus ice clouds are based on environmental conditions such as synoptic weather systems, orographic lifting, and convective
systems (Muhlbauer et al., 2014; Jackson et al., 2015). Independent of these classifications, the life cycle of cirrus clouds
consists of a formation and growth phase under influence of freezing or nucleation and deposition, followed by decay and
disappearance of ice due to fall out (sedimentation) and sublimation of ice particles. These processes have been clearly

identified in the vertical structure of ice clouds based on observations and high-resolution model simulations (Urbanek et al.,
2017; Gasparini et al., 2019), and in Lagrangian studies considering the cirrus life-cycle along trajectories (Gasparini et al.,
2021).

Stable water isotopes are a useful natural tracer of the Earth's water cycle (Galewsky et al., 2016), and in particular for studying

processes near tropical deep convection (Bony et al., 2008; Blossey et al., 2010). Besides the most abundant $H_2^{16}O$
isotopologue, the Earth's water cycle also contains the much less abundant heavier HDO and $H_2^{18}O$ isotopologues (Gat, 1996).
Differences in their physical characteristics (saturation vapour pressure and molecular diffusivity) change their relative
proportions during phase transitions, referred to as fractionation, whereby the heavier isotopes go preferentially into the solid
or liquid phase as compared to the vapour phase (Merlivat and Nief, 1967; Jouzel and Merlivat, 1984). As a result, isotopes

contain information on an air parcel's history, making it a unique tool to study moist atmospheric processes, including
precipitation, convection and synoptic weather systems (Bony et al., 2008; Risi et al., 2008a, 2010a; Pfahl et al., 2012;
Aemisegger et al., 2015; Weng et al., 2021). An important concept that helps interpreting isotope information is Rayleigh
distillation, which describes the progressive depletion of heavy isotopes in vapour of cooling (e.g. moist-adiabatically
ascending) air typically assuming that the forming condensate is instantly removed (Dansgaard, 1964). This principle explains,

to a first order, the observed depletion of heavy isotopes in water vapour with latitude and altitude (Dansgaard, 1964; Rozanski
et al., 1993). In the tropics, these effects are less prominent due to the relatively weak horizontal temperature gradients, and
strong convective transport and mixing that can dominate over horizontal transport. In these regions, there is often an
anticorrelation observed between heavy isotopes in precipitation and precipitation amounts (Dansgaard, 1964; Rozanski et al.,
1993). This effect can largely be explained by the strength of convection (Risi et al., 2020) and its degree of organisation

(Lawrence et al., 2004, Nlend et al., 2020). Stronger convection usually has larger rain droplets and occurs in a moister
environment, reducing the reevaporation of the falling condensate and leading to more depleted surface precipitation (Risi et
al., 2008b, Torri et al., 2017). Similarly, strong, organised convection typically entrains more depleted vapor from higher
altitudes and more efficiently depletes low tropospheric vapour than weak, isolated convection, eventually resulting in more
depleted surface precipitation (Risi et al., 2008a; Moore et al., 2014).


Observations of water isotopes and modelling experiments equipped with isotope physics have improved our understanding
of atmospheric processes in monsoons and tropical convection in general. Global satellite-based measurements and GCM
simulations showed that ocean evaporation and continental evapotranspiration provide enriched vapour that is transported into



monsoon systems (Worden et al., 2007; Risi et al., 2010a; Brown et al., 2013). Land-inward moving air is gradually depleted
due to rainout along its transport pathways, partly compensated by continental moisture recycling (Risi et al., 2010a; Winnick
et al., 2014). Large-scale subsidence can dehydrate and deplete vapour at the poleward edges of monsoon systems
(Frankenberg et al., 2009, Risi et al., 2010a). Convective processes and their influence on water isotopes are more complex
than the effects of large-scale circulation processes and limit our understanding and interpretation of isotope signals in
atmospheric water. Satellite measurements suggest that shallow convection enriches mid-tropospheric vapour through
detrainment of enriched boundary layer air (Lacour et al., 2018), whereas deep convection enriches upper tropospheric air due
to convective detrainment at these altitudes (Randel et al., 2012). Experiments with a single column model pointed to a variety
of processes that can affect the isotopic composition of water vapour and precipitation in relation to tropical deep convection,
including condensate lofting, rain reevaporation, unsaturated downdrafts, and the isotopic exchange between droplets and the
tropospheric environment (Bony et al., 2008; Risi et al., 2008b). Specifically, with regards to the WAM, in situ observations
of precipitation isotopes have shown that its isotopic composition can reflect information about the type of convective
organisation (Risi et al., 2008a, 2010b).

Importantly, observations of stable water isotopes brought novel insights into convective processes that affect the TTL and its
water budget. The TTL demarcates the transition of the upper troposphere to the lower stratosphere in the tropics between
altitudes of 14 and 18.5 km, corresponding to approximately 150 and 70 hPa (Fueglistaler et al., 2009). This region also
contains the cold point tropopause with temperatures as low as 190 K near 90 hPa and 17 km altitude (Fueglistaler et al., 2009).
Transport of air through this cold point leads to so-called "freeze-drying" of air as a result of in situ formation of ice that
subsequently falls out (Holton and Gettelman, 2001), further depleting the remaining vapour from heavy isotopes. Previously,
it was thought that slow upwelling of air through this cold point, associated with the Brewer-Dobson circulation, controls the
very low amounts of water vapour that can enter the lower stratosphere (Holton et al., 1995). Spaceborne observations of stable
water isotopes, however, showed that tropical lower-stratospheric water vapour is much more enriched than Rayleigh
distillation predicts, suggesting other processes at work such as tropical deep convection (Moyer et al., 1996; Kuang et al.,
2003; Hanisco et al., 2007; Steinwagner et al., 2010). Indeed, subsequent in situ observations from aircraft measurements
(Corti et al., 2008; Sayres et al., 2010) and modelling experiments (Ren et al., 2007; Wang et al., 2019; Bolot and Fueglistaler,
2021) showed that evaporation of convectively lofted ice moistens the lower TTL. Moreover, measurements over West Africa
suggest that occasionally overshooting convection can directly moisten the lower stratosphere (Khaykin et al., 2009). A variety
of model experiments equipped with stable water isotopes confirmed the influence of tropical convection on the TTL water
budget, such as conceptual models (Dessler and Sherwood, 2003), trajectory calculations (Dessler et al., 2007; Sayres et al.,
2010), single column models (Bony et al., 2008), large eddy simulations (Smith et al., 2006), and global circulation model
simulations (Eichinger et al., 2015). Idealized simulations with a two-dimensional cloud-resolving model suggested that
sublimation of convectively lofted enriched ice and fractionation of in situ cirrus cloud formation affects the water vapour in
the TTL (Blossey et al., 2010). Accordingly, previous modelling studies that investigated the role of deep convection and ice



cloud formation on the TTL water budget used idealized model setups, trajectory calculations with inherent limitations stemming from the underlying data, or relatively coarse resolution model simulations with parameterised convection.


This study uses, for the first time, regional isotope-enabled convection-permitting model simulations constrained by actual meteorological conditions to investigate the atmospheric processes related to tropical deep convection and cirrus clouds in the WAM. The purpose is twofold: (1) to identify the processes related to tropical ice cloud formation and decay based on stable water isotope information, and (2) to quantify the impact of deep convection on the water budget of the TTL and its isotopic

composition. We use simulations with the COSMO$_{iso}$ model (Pfahl et al., 2012) with different resolutions and treatments of convection over West Africa for the period of June-July 2016 (Sect. 2). First, we evaluate the credibility of the simulations by comparing monthly precipitation isotopes to observations. In addition, we compare the precipitation characteristics from the best performing simulation to insights from the DACCIWA field campaign (Knippertz et al., 2017) that took place during that period (Sect. 3). After confirming realistic behaviour of convective storms in our simulations, we switch focus to tropical ice

clouds, and explore the isotopic imprints of deep convection on the atmosphere in a detailed case study, followed by a statistical analysis (Sect. 4). Furthermore, we use total column ice amounts as a simple proxy to distinguish between different types of ice clouds and examine differences in their isotope signals (Sect. 5). This approach also supports a quantitative assessment of the impact of deep convection on the TTL water budget and its isotopic composition. Finally, Sect. 6 concludes this study with a summary of the results, including a synthesis of five different key processes related to the formation and decay of tropical

ice clouds derived from isotope information.

## 2 Methods and Data

### 2.1 The COSMO$_{iso}$ model

The Consortium for Small-Scale Modelling (COSMO) model (Steppeler et al., 2003) is a non-hydrostatic limited-area weather and climate model used for operational forecasting and research purposes. Stable water isotopes HDO and H$_2^{18}$O are

implemented in the model (COSMO$_{iso}$) through adding parallel water cycles for both heavy isotopes that are purely diagnostic and do not affect any other model components (Pfahl et al., 2012). These heavy isotopes undergo identical processes as the common water isotope H$_2^{16}$O except during phase transitions when isotopic fractionation occurs. COSMO$_{iso}$ has been used to study a variety of atmospheric processes in the hydrological cycle in idealized studies (Dütsch et al., 2016), detailed case studies (Pfahl et al., 2012; Aemisegger et al., 2015; Lee et al., 2019), and (semi-)climatological analyses (Dütsch et al., 2018;

Christner et al., 2018; Dahinden et al., 2021; Diekmann et al., 2021; Thurnherr et al., 2021).

This study uses the isotope-enabled COSMO model version 4.18 with a prognostic one-moment bulk microphysics scheme (Doms et al., 2005). Water is represented by five species: vapour, cloud ice, snow, cloud water, and rain. Cloud water and cloud ice represent small droplets and pristine ice crystals with negligible fall velocities. Rain and snow represent larger





droplets and ice particles, respectively, with fall velocities that are computed from size distribution functions and empirically
       derived relations based on measurements (Doms et al., 2005).

       Fractionation occurs during the phase changes from vapour to liquid (condensation), vapour to ice (nucleation and deposition),
       and liquid to vapour (evaporation). Fractionation is parameterized using equilibrium fractionation factors with respect to liquid
water and ice, following Majoube (1971) and Merlivat and Nief (1967), respectively. Non-equilibrium fractionation effects
       occur, for instance, if the air is supersaturated with respect to ice, which is taken into account by a combined fractionation
       factor (Jouzel and Merlivat, 1984, Blossey et al., 2010). During freezing, melting and sublimation no fractionation occurs. For
       more information on the representation of fractionation during phase changes in the cloud microphysics scheme of COSMO$_{iso}$,
       see Pfahl et al. (2012).


       Surface evaporation from ocean is parameterized using the Craig-Gordon model (Craig-Gordon, 1965) representing non-
       equilibrium fractionation based on an empirically-derived relation independent of wind speed (Pfahl and Wernli, 2009). Land-
       atmosphere interactions are represented by a multilayer land surface scheme equipped with water isotopes, accounting for
       processes such as plant transpiration and bare soil evaporation, see for more details the supplement of Christner et al. (2018).

**2.2 Model simulations**

       The COSMO$_{iso}$ model simulations are performed on a rotated grid centred at 7.5°E and 13.0°N. The model domain covers the
       larger part of the African continent, the eastern and tropical Atlantic, the Mediterranean, and the wester margins of the Indian
       Ocean (Fig. 1). This relatively large model domain allows the explicit simulation of different inflow airstreams into the WAM
       system and the processes that affect the isotopic composition of water along these transport pathways (Diekmann et al., 2021).
We perform three model simulations with different configurations in terms of their resolution and treatment of convection
       (Table 1). The first simulation, PAR14, has a horizontal grid spacing of 14 km, 40 vertical model levels and convection is
       parameterized using the Tiedtke scheme (Tiedtke, 1989). The second simulation, EXPL14, has an identical resolution, but the
       convection parameterization scheme is switched off. The third simulation, EXPL7, is also in convection-permitting setup (i.e.
       no parameterized convection), but at a higher resolution with a 7 km horizontal grid spacing and 60 model levels reaching up
to 23.6 km a.s.l. For both simulations in convection-permitting setup, also the shallow convection scheme is switched off.

       The motivation for these different model experiments is that it is not a priori clear how the isotope-enabled model performs in
       convection-permitting setup for the WAM. Most previous studies used COSMO$_{iso}$ in mid-latitude or polar regions with
       parameterized convection. For our present study it is highly advantageous to switch off the convection scheme, allowing the
explicit simulation of deep moist convection in the tropics. This model setup will benefit the interpretation of isotope signals
       in ice clouds, circumventing the drawbacks of the simplified cloud microphysics in the convection parameterization scheme
       and the complex interactions of convection and cloud microphysics schemes in cirrus layers. Moreover, previous studies have



found that the hydrological cycle and deep moist convection in the WAM is much better represented in convection-permitting simulations than with parameterized convection, even at relatively coarse resolutions (Marsham et al., 2013; Birch et al., 2014).
Our choice for relatively coarse-resolution simulations considering the usual standards for a convection-permitting setup is further motivated by new insights that parameterization schemes can be switched off at coarser resolutions than previously thought, that is, on the order of ~10 km (Vergara-Temprado et al., 2020). This resolution also allows the use of a large model domain in which the regional model can generate its own hydrological cycle and isotope meteorology, without being too strongly affected by the boundary information from the driving global model output.


Our COSMO$_{iso}$ simulations are initialized and driven by 6-hourly updated fields at the lateral boundaries from global ECHAM5-wiso output (Werner et al., 2011). This output is produced by global isotope-enabled ECHAM5 simulations with a T106 spectral resolution and 31 model levels, nudged to the dry atmospheric state of ERA-Interim reanalysis (Butzin et al., 2014). The COSMO$_{iso}$ simulations are relaxed towards the driving global output using a spectral nudging (von Storch et al.,
2000) of the large-scale horizontal winds above 850 hPa (wave numbers 5 and less). In this way, the large-scale circulation within the regional domain is kept close to real conditions while smaller-scale processes, including deep moist convection, are fully treated by COSMO$_{iso}$. The COSMO$_{iso}$ model simulations are integrated for the 2-month period of the DACCIWA field campaign from 1 June to 30 July 2016. Hourly output is produced on model levels and interpolated to 29 pressure levels with 50 hPa intervals between 750 to 250 hPa, and 25 hPa intervals in the 1000 to 750 and 250 to 50 hPa pressure layers.

**2.3 Observations**

For the purpose of model validation, we compare isotope signals in precipitation from the different COSMO$_{iso}$ simulations to monthly observations from the Global Network for Isotopes in Precipitation (GNIP). GNIP data are collected by the International Atomic Energy Agency (IAEA) and WMO (Rozanski et al., 1993; Araguas-Araguas et al., 2000). More specifically, we use monthly observations from 19 stations over Equatorial and North Africa. Several stations record only $\delta$
values of $^{18}O$ and $^{2}H$ in precipitation and miss the actual precipitation amounts. For this reason, we refrain from computing mass-weighted means over the two-month simulation period and we compare the model output to observations for the two individual months (see Sect. 3.1). Note that the comparison of the model output to observations provides a relatively simplistic evaluation of the model simulations with the intention to explore the credibility of the simulations rather than performing a profound model validation for which simulations over longer time periods and more observational data would be required.



## 3 Precipitation

### 3.1 Comparison of COSMO$_{iso}$ to GNIP observations

Figure 2 shows monthly precipitation and its isotopic composition ($\delta^2$H and deuterium excess, $d$-excess) for the three model simulations and the GNIP observations in July 2016 (for June 2016, see Fig. S1), where $d$-excess is defined as $d = \delta^2$H $- 8\,\delta^{18}$O, which provides a measure for non-equilibrium fractionation processes (Craig, 1961; Dansgaard, 1964). The joint evaluation of these three quantities (precipitation amounts, $\delta^2$H and $d$) provides an overall assessment of the representation of precipitation processes in the model simulations. We first discuss the spatial patterns in precipitation isotope signals in the three simulations, and then compare them to GNIP observations.

At a first glance precipitation generally displays a similar pattern in the three model simulations. Precipitation occurs in a zonal band between 0-20°N that reflects the tropical rain belt as a part of the Inter-Tropical Convergence Zone (Fig. 2a,d,g). $\delta^2$H in precipitation is generally more depleted in the centre of the rain belt where rainfall amounts are highest (Fig. 2b,e,h), consistent with previous studies about isotopes in tropical precipitation (e.g. Dansgaard, 1964; Rozanski et al., 1993; Risi et al., 2008b).

Upon closer scrutiny, we note substantial spatial gradients in precipitation isotope signals. They are particularly visible in the convection-permitting simulations (Fig. 2e,h) and reflect three different processes: (1) enriched precipitation over the coast of Guinea that depletes towards the Sahel region due to the gradual depletion of land-inward moving moist air from the Tropical Atlantic, although partly compensated by continental recycling (Risi et al., 2010a; Winnick et a., 2014), (2) very enriched precipitation over Central Africa, likely due to the influence of non-fractionating transpiration in this densely vegetated region (Galewsky et al., 2016), and (3) very enriched precipitation along the northern fringe of the rain belt as a result of strong below-cloud effects, i.e., enhanced evaporation of falling rain in the very warm and dry environment (Risi et al., 2008a). Signals of $d$-excess in precipitation in these three regions (Fig. 2f,i) are generally consistent with the same processes as above: (1) decreasing $d$-excess along air parcels moving land inward due to the weak but non-negligible impact of the condensation temperature on the $d$-excess (Thurnherr et al., 2021, their Appendix A), (2) $d$-excess tends to be lower when the share of transpiration (non-fractionating) in evapotranspiration is relatively large (Risi et al., 2010a; Aemisegger et al., 2014), which can reasonably be assumed over central Africa, and (3) very low $d$-excess along the northern fringe of the tropical rain belt due to strong non-equilibrium fractionation during below cloud evaporation in dry air (Araguas-Araguas et al., 2000; Risi et al., 2008a; Graf et al., 2019).

Interestingly, the third process, leading to low $d$-excess values due to strong rain evaporation along the northern fringe of the rain belt, is not observed in the parameterized convection simulation (Fig. 2c). Generally, precipitation in the parameterized convection simulation has larger amounts, is more depleted and has higher values of $d$-excess compared to both convection-permitting simulations (Fig. 2a-i). Most likely, these differences stem from how precipitation processes (convective and large-





scale components) are represented in the different model configurations, but it is beyond the scope of this study to address this in more detail. Relevant here is that precipitation and its isotopic composition clearly differ between parameterized and explicit

convection simulations, while the differences between the coarser and higher resolution convection-permitting simulations are much smaller. Consistent with previous studies, switching off the parameterization scheme strongly changes the hydrological cycle in the WAM region (Marsham et al., 2013; Birch et al., 2014; Pante and Knippertz, 2017).

GNIP precipitation and isotope information follow a similar spatial distribution as the output from the COSMO$_{iso}$ simulations

(Fig. 2j-m). Precipitation is relatively enriched over the coast of Guinea and Central Africa and more depleted over the Sahel (Fig. 2k). For a quantitative comparison of the simulations to observations (Fig. 2m-o), we interpolate the COSMO$_{iso}$ output to the station locations. Overall, we observe that the COSMO$_{iso}$ simulations follow the GNIP values reasonably well. A few interesting deviations appear that can be clarified by physical arguments. For example, for a station in Uganda (UG2), the convection-permitting simulations underestimate the precipitation amounts, going along with too high $\delta^2$H and too low $d$-

excess in precipitation as compared to GNIP (Fig. 2m-o). This suggests too much below-cloud rain evaporation, too shallow clouds, or too low precipitation efficiency in these simulations at this location, consistent with too enriched surface precipitation. In contrast, for the station in Ethiopia (ET) all three model simulations overestimate rainfall amounts and simulate too low $\delta^2$H compared to observations, showing that biases in simulated precipitation amounts are consistent with biases in $\delta^2$H in precipitation. Likewise, in June 2016 (Fig. S1m-o), the simulations overestimate precipitation amounts and have too

depleted precipitation as compared to the GNIP observations at both stations in Uganda (UG2) and Ethiopia (ET).

Based on the GNIP observations and model output interpolated at station locations, we compute the root mean square error (RMSE) and mean error of the model simulations as compared to the observations (Tables 2 and 3). The RMSE of the isotope variables ($\delta^{18}$O, $\delta^2$H and $d$) is lowest for the EXPL7 simulation in both months, indicating that the convection-permitting

simulation at 7 km performs better than both simulations at 14 km. At this resolution, there is no clear difference between the parameterized and explicit convection setup as EXLP14 performs better than PAR14 in June, while no clear differences between both simulations emerge in July (Table 2). Mean errors show that the convection-parameterized simulation consistently produces too depleted precipitation, as also found in a 30-year climatological study for central Europe (Dütsch et al., 2018). In contrast, the convection-permitting simulation at 14 km produces mostly too enriched surface precipitation.


This brief model evaluation demonstrates that the COSMO$_{iso}$ simulations are overall in good agreement with the GNIP observations. Moreover, the convection-permitting simulation at 7 km outperforms both simulations at 14 km, and will be used for the remaining part of this study to analyse tropical ice clouds in the WAM of summer 2016. Before switching focus to tropical ice clouds, we discuss the evolution of precipitation during the monsoon period and the motion of convective storms

in the EXPL7 simulation.



## 3.2 Time evolution

The period of June-July 2016, when the DACCIWA field campaign took place, was characterized by four phases in the WAM evolution: (1) the pre-onset until 21 June, (2) the post-onset from 22 June-20 July, (3) a wet phase from 21-26 July, and (4) a return to undisturbed monsoon conditions from 27-31 July (Knippertz et al., 2017). These four phases are reasonably well

represented in the EXPL7 simulation in terms of precipitation amounts and its latitudinal distribution (Fig. 3a,b). Before the monsoon onset (phase 1), precipitation is predominantly confined to lower latitudes across the coast of Guinea (~0-7.5°N). After the monsoon onset (phase 2), the rain belt has migrated northward and is positioned over the Sahel region (7.5-15°N). During the wet episode from 21 to 26 July (phase 3), large precipitation amounts occur over both the coast of Guinea and the Sahel, after which more usual monsoon conditions follow until the end of July (phase 4) with precipitation primarily centred

over the Sahel. This evolution is fully in line with the observation-based description in Knippertz et al. (2017), their Figs. 5 and 6, further supporting the credibility of EXPL7. Our model simulates the monsoon onset two days too early (19-20 June), but captures very well the abrupt northward shift of precipitation into the Sahel (Fig. 3b). Precipitation is most depleted during periods of enhanced precipitation in mid-June and during phase 3 (Fig. 3a), which is consistent with previous studies that linked precipitation amounts to its isotopic composition (Risi et al., 2008a), but cannot be compared to observations as these

are not available at these timescales. Figure 3c shows a Hovmöller diagram with average precipitation over the latitude band 5-20°N as a function of time and longitude (Fig. 3c). This clearly demonstrates long-lived MCSs that move from east to west over the course of several days, consistent with Knippertz et al. (2017), their Fig. 10, providing further confidence in the simulation of monsoon MCSs in EXLP7. On 15 June 2016, an intense MCS passed our region of interest, as reflected by large precipitation amounts on this day (see the markers in Fig. 3b,c). This specific day is chosen for a case study to explore the

effects of tropical deep convection on ice clouds and their isotope signals.

## 4 Tropical ice clouds and their isotope signals

### 4.1 An MCS case study

Figure 4 shows the spatial distribution of ice and its isotopic composition at 200 hPa at 18:00 UTC on 15 June 2016. Ice content is here and throughout this manuscript, unless explicitly mentioned otherwise, defined by the sum of cloud ice and snow (Fig.

4a). Similarly, $\delta^2$H in ice refers to the $\delta$ values in the mass-weighted sum of cloud ice and snow (Fig. 4b). Figure 4c shows the deviation of the isotopic composition of ice ($\delta^2$H$_{ice}$ as in Fig. 4b) from the ice that would from local vapour under equilibrium fractionation. This variable, hereafter referred to as **disequilibrium in ice**, is defined by: $\Delta\delta^2$H$_{ice} = \delta^2$H$_{ice} - \delta^2$H$_{ice,eq}$ (1), whereby equilibrium ice $\delta^2$H$_{ice,eq}$ is temperature dependent and computed using the same fractionation factor as in COSMO$_{iso}$, $\alpha = e^{(16288/T^2 - 0.0934)}$, where $T$ is the temperature (Merlivat and Nief, 1967; Rowley and Garzione, 2007).

This approach follows the rationale of Aemisegger et al. (2015) and Graf et al. (2019), who used the disequilibrium of



precipitation isotopes from vapour isotopes to investigate the influence of below-cloud effects during the passage of cold fronts in Europe.

Across Equatorial Africa we note multiple convective systems that reach the upper troposphere, characterised by large ice content > 1000 mg kg$^{-1}$ (Fig. 4a). The ice is very enriched with $\delta^2$H values near –100 ‰ (Fig. 4b), indicating that the convective systems bring large amounts of very enriched ice into the upper troposphere, which has formed at low altitudes from enriched vapour deposition or through freezing liquid. This is confirmed by large positive $\Delta\delta^2$H$_{ice}$ values that exceed +300 ‰ within these convective systems (Fig. 4c). Farther away from the convective systems we observe much lower amounts of ice (< 25 mg kg$^{-1}$), typical of cirrus clouds, which is much more depleted ($\sim$ –600 to –350 ‰), and has near zero to negative

$\Delta\delta^2$H$_{ice}$ values. Neutral $\Delta\delta^2$H$_{ice}$ is consistent with the in situ formation of ice from local vapour under equilibrium fractionation, while negative $\Delta\delta^2$H$_{ice}$ likely reflects the sedimentation of ice from aloft, being relatively depleted with respect to the ambient vapour at these altitudes.

  Figure 5 zooms in at an MCS in the centre of the model domain (see boxes in Fig. 4a-c) at 16:00 UTC on 15 June 2016. In

addition to Fig. 4, Fig. 5 also shows the $\delta^2$H in vapour (Fig. 5c), vertical cross sections in the zonal direction (Fig. 5e-h), and vertical motion contours. In the zonal cross sections, the liquid, mixed-phase and ice cloud layers are denoted by the light grey contours (at $T = 0$°C and –38°C), and the approximate cold point tropopause is demarcated by dark grey contours (Fig. 5e-h). Based on this figure and its time sequence (not shown), we observe how convective updrafts within the MCS inject enriched ice into the upper troposphere. This enriched ice remains for several hours in the middle and upper troposphere on the eastern

flank of the westward moving storm and in its vicinity. Interestingly, ice is very enriched in both the updrafts and downdrafts of the storm (Fig. 5b,d,f,h), whereas the vapour is very depleted in the upper part of the updrafts and very enriched within the downdrafts of the storm, reaching below –768 ‰ and over –448 ‰, respectively, at altitudes near 125 hPa (Fig. 5g). This near factor two difference in $\delta$ values of vapour isotopes suggests a very strong influence of deep moist convection on vapour in the upper troposphere and the lower TTL. Intense condensation and deposition within the updrafts deplete the vapour from

heavy isotopes that preferentially go into the ice phase, whereas non-fractionating sublimation of very enriched ice in the downdrafts enriches the ambient vapour of the upper troposphere. Particularly noteworthy are the rings of enriched vapour around the convective cloud shields that can be explained by the same processes in the horizontal outflow of convective storms (Fig. 5c). Furthermore, we observe negative $\Delta\delta^2$H$_{ice}$ values in the mixed-phase cloud layer (between temperatures of 0°C and –38°C) in the convective cloud region as well as in the ice layer ($T < $–38°C) of the cirrus shield, reflecting the sedimentation

and sublimation of ice that formed at higher altitudes in these clouds (Fig. 5h). It is also worth to note the marginally negative $\Delta\delta^2$H$_{ice}$ values in the lower parts of the convective updrafts, where liquid hydrometeors start to freeze. Freezing of liquid water is a non-fractionation process, while condensation has a lower equilibrium fractionation factor than the direct transition from vapour to ice, explaining the slightly negative $\Delta\delta^2$H$_{ice}$ values in ice at the base of convective updrafts.



Based on the analysis above, we distinguish the following five key processes related to tropical ice clouds as derived from isotope information, and in particular the disequilibrium in ice (Fig. 5h): (1) **convective lofting of enriched ice** into the upper troposphere with very positive $\Delta\delta^2H_{ice}$ values (> 300 ‰), (2) **in situ ice formation under equilibrium fractionation** with near zero $\Delta\delta^2H_{ice}$ values, (3) **sedimentation and sublimation of ice** in the mixed-phase cloud layer of convective systems and the lower parts of the cirrus shields with relatively large negative $\Delta\delta^2H_{ice}$ values (down to ~ –200 ‰), (4) non-

fractionating **sublimation of ice in convective downdrafts that enriches the environmental vapour,** reflected by moderately negative $\Delta\delta^2H_{ice}$ values in the mixed-phase cloud layer (near –100 ‰), and (5) the **freezing of liquid water** in the lower parts of convective updrafts in the mixed-phase cloud layer with small negative $\Delta\delta^2H_{ice}$ values (~ –25 ‰). These five processes are indicated by the corresponding labels in Fig. 5h and will be further discussed and quantified in a statistical analysis in Sect. 4.2. Appendix A presents detailed information on the interpretation of disequilibrium in ice in light of these five processes.


The isotope signatures of the MCS on 15 June 2016 are not exceptional, but typical of convective storms in our WAM simulation. Figure 6 shows the time evolution of vertical profiles of ice content, $\delta^2H$ in ice, and disequilibrium in ice for a small 1°x1° region (black boxes in Fig. 5a-d) throughout the simulation period of June-July 2016. Multiple convective storms pass through this small region, as revealed by large amounts of ice content that reaches into the upper troposphere (Fig. 6a).

Each of these convective storms is characterized by high values of $\delta^2H$ in ice and very positive $\Delta\delta^2H_{ice}$ values (Fig. 6b,c). Intermittent periods between convective storms indicate the presence of cirrus clouds with lower ice content, more depleted ice, and near zero $\Delta\delta^2H_{ice}$ values in the centre of these clouds at ~150 hPa and negative $\Delta\delta^2H_{ice}$ values in the lower parts of these clouds (Fig. 6a-c). $\delta^2H$ in vapour does not exhibit a clear signal during passages of MCSs due to the spatial averaging over both updrafts and downdrafts within MCSs that concur over the 1° target region (not shown).

## 4.2 Statistical distribution of water species and their isotope signals

In this section, we proceed to a statistical analysis of water and its isotopic composition in the WAM region of interest (10°W-20°E, 5°-20°N, see the black box in Fig. 1) in July 2016. Figure 7a shows the vertical profiles of various water species horizontally averaged in the region. Note that we consider here cloud ice and snow separately as an exception to all other analyses in this study. Cloud ice content reaches a maximum of ~ 4.4 mg kg$^{-1}$ near 250 hPa, while snow content attains an

about tenfold larger maximum of ~ 70 mg kg$^{-1}$ near 300 hPa. Importantly, the fraction of summed cloud ice and snow in total water is largest between 100-200 hPa and exceeds 50% in this layer. Figure 7b shows the corresponding vertical profiles of $\delta^2H$ values in vapour, cloud ice, and snow. $\delta^2H$ in vapour (black line) decreases with height approximately following Rayleigh distillation until an altitude near 400 hPa and transits above away from the Rayleigh regime with increasing $\delta$ values with height for $p < 150$ hPa, consistent with previous model studies and observations (Kuang et al., 2003; Bony et al., 2008; Randel

et al., 2012). The isotopic composition of vapour reaches a minimum near the bottom of the TTL at 150 hPa, well below the cold point tropopause near 90 hPa (Fig. 7a), with $\delta$ values near –590 ‰. This value is roughly in agreement with the –650 ‰



estimates from the Atmospheric Trace Molecule Spectroscopy (ATMOS) measurements in November 1994 (Kuang et al., 2003), the –650 ‰ from satellite-based measurements across the tropics (Randel et al., 2012), minimum values between –600 and –400 ‰ from aircraft measurements in the North American monsoon (Hanisco et al., 2007), and the –530 ‰ from single column model simulations in a tropical environment (Bony et al., 2008). Ice that would form under equilibrium fractionation from local vapour (turquoise line) demonstrates a very similar profile as $\delta^2H$ in cloud ice from the model output (blue line; Fig. 7b). These very similar vertical profiles reflect the in situ formation of small ice crystals with negligible fall velocities. In contrast, the vertical profile of $\delta^2H$ in snow (red line; Fig. 7b) deviates substantially and shows roughly constant $\delta$ values with height near –150 ‰. We hypothesise that this profile results from strong vertical transports of the frozen condensate in convective regions.

To provide a more detailed picture of the isotopic composition of the water species, we consider their mass-based probability density functions at three single pressure levels (solid lines in Fig. 7c-e). Consistent with our earlier note, at all three levels the distributions of $\delta^2H$ in cloud ice from model output are close to the local equilibrium values, illustrating the in situ formations of small, suspended ice crystals under equilibrium fractionation. Interestingly, the mass-based probability density function of $\delta^2H$ in snow (red solid lines in Fig. 7c-e) shows a bimodal distribution at 125 and 200 hPa with a primary peak near –130 ‰ and a secondary peak near –425 ‰ hPa. The probability density functions based on occurrence frequencies (red dashed lines in Fig. 7c-e) show that the enriched peak results from relatively few occurrences, in contrast to the depleted peak that corresponds to very large occurrence frequencies. These distributions suggest that the convective lofting of large amounts of enriched snow occurs in sparse updrafts, while much lower amounts of more depleted snow forms through aggregation of in situ ice formed crystals in widespread cirrus. At 450 hPa, both the mass and frequency-based distributions of snow are shifted towards more depleted $\delta^2H$ values compared to equilibrium ice, as a result of the sedimentation of relatively depleted snow from higher altitudes (Fig. 7e).

Consistent with the case study, this analysis highlights two different formation pathways of ice that we will further investigate. Figure 8 shows the frequency distribution of ice and its isotopic composition on pressure levels from 50 to 700 hPa with overlaid the average vertical motion for each individual bin. For clarity, here we consider again the sum of cloud ice and snow. Ice clouds occur most frequently in the upper troposphere (125-250 hPa) with water content between 0.2 and 25 mg kg$^{-1}$ (Fig. 8a). Most ice gradually depletes with altitude (Fig. 8b). Ice in this regime forms in situ under equilibrium fractionation as indicated by the near zero disequilibrium in ice (Fig. 8c), especially in the upper troposphere (100-200 hPa). These high frequencies in the distribution are suggestive of depositional growth and aggregation of in situ formed ice crystals in vast cirrus shields. Interestingly, a part of the ice cloud distribution differs remarkably from this regime. Few regions with relatively low occurrence frequencies have very large amounts of ice content (> 2000 mg kg$^{-1}$) associated with strong convective updrafts with an average upward motion > 1 m s$^{-1}$ (Fig. 8a). This part of the distribution also emerges as relatively enriched ice for $p <$ 400 hPa that clearly deviates from the in situ formation regime, indicative of convective lofting of enriched ice (Fig. 8b).





Indeed, the distribution of the disequilibrium in ice demonstrates occurrences with increasingly larger positive $\Delta\delta^2H_{ice}$ values with height, exceeding +600 ‰ in the lower TTL (100-125hPa; Fig. 8c). These positive $\Delta\delta^2H_{ice}$ values coincide with strong upward motion in the lower parts of these positive values. Thus, the frequency distribution in Fig. 8 demonstrates convectively lofted enriched ice and in situ formed ice under equilibrium fractionation (labels "1" and "2" in Fig. 8, respectively). These

results show large similarities to those from Blossey et al. (2010), their Fig. 6. In addition, Fig. 8c also shows that a substantial part of the distribution falls in the range of negative $\Delta\delta^2H_{ice}$ values near ~ –200 ‰ and even down to ~ –360 ‰. These signals stem from sedimenting ice that formed at higher altitudes and is relatively depleted with respect to local vapour conditions. These negative $\Delta\delta^2H_{ice}$ values occur from the upper troposphere to the melting level (200-600 hPa) and thus include sedimentation of ice from both cirrus clouds in the upper troposphere and convective clouds in the mixed-phase cloud layer.

Interestingly, we also observe weak downward motion in the 300-600 hPa pressure layer for moderately negative $\Delta\delta^2H_{ice}$ values between –40 and –120 ‰, reflecting the signatures of convective downdrafts. The next section systematically investigates the relationship between vertical motion and the isotope signals in ice and water vapour to further identify the relevant processes in convective updrafts and downdrafts.

### 4.3 Convective updrafts and downdrafts

Figure 9 presents the characteristics of water species and their isotopic compositions as a function of vertical motion across pressure levels. For each bin, we retrieve the occurrence frequencies, average ice content, $\delta^2H$ in ice, $\delta^2H$ in vapour, and the disequilibrium in ice (Fig. 9a-e). In addition, Fig. 9f shows the average relative humidity over ice (RH$_{ice}$), which is computed from temperature, pressure and specific humidity on model levels, and then interpolated on pressure levels to minimize the impact from interpolation errors. Not surprisingly, the distribution is centred near zero vertical motion and displays some

asymmetry with slightly enhanced frequencies of weak downward motion and an extended 'tail' towards very strong upward motion (Fig. 9a). Both strong updrafts and downdrafts contain very large amounts of ice that is relatively enriched compared to more depleted ice in regions with very weak vertical motion (Fig. 9b,c). The disequilibrium in ice exceeds 500 ‰ in the updrafts and 300 ‰ in weaker downdrafts near 125-150 hPa (Fig. 9e). Interestingly, the isotopic composition of vapour at these altitudes is very depleted in strong updrafts ($\delta^2H < –750$ ‰) and very enriched in downdrafts ($\delta^2H \sim –400$ ‰) at slightly

lower altitudes near 175 hPa (Fig. 9d). This large range in $\delta^2H$ in vapour correspond to our earlier findings based on the MCS case study (Sect. 4) and is here corroborated by the month-long statistical analysis. Moreover, this large range of $\delta^2H$ in vapour is consistent with observations (Webster and Heymsfield, 2003; Hanisco et al., 2007; Sayres et al., 2010). We provide here evidence that these large fluctuations are directly related to vertical motion and the contrasting processes that take place in the convective updrafts and downdrafts.


The information in Fig. 9 also serves perfectly to illustrate the five key processes postulated in Sect. 4.1, see the labels 1-5 in Fig. 9. We briefly reiterate these five processes based on the statistical analysis: (1) Convective ice lofting corresponds to the



right-hand-side of the distribution (i.e. strong updrafts) and is characterised by large amounts of ice content (> 5000 mg kg$^{-1}$), relatively high $\delta^2 H_{ice}$ values ($\sim$ –100 ‰), very large positive $\Delta\delta^2 H_{ice}$ values (> 500 ‰), and very low $\delta^2 H_{vap}$ values (< –768

‰) as a result of the preferential condensation and deposition of heavy isotopes in these updrafts, while (2) in situ formation of ice under equilibrium fractionation in regions with weak or negligible vertical motion has moderate ice content (0.1-10 mg kg$^{-1}$), relatively depleted ice ($\sim$ –400 ‰), and near zero $\Delta\delta^2 H_{ice}$ values. (3) In lower parts of the troposphere with weak vertical motion, we note the signatures of sedimenting and sublimating ice that formed at higher altitudes and is relatively depleted with respect to the local vapour conditions, reflected by moderately negative $\Delta\delta^2 H_{ice}$ values near –100 ‰, especially near the

melting level at $\sim$ 600 hPa. (4) In upper tropospheric regions of strong convective downdrafts on the left-hand-side of the distribution, sublimation of enriched ice enriches the ambient vapour with $\delta^2 H$ values that are almost twice as high as in regions of convective updrafts at similar altitudes. Sublimation in downdrafts is likely to occur as the RH$_{ice}$ falls below 100 % (Fig. 9f). (5) In mid-tropospheric regions of moderate upward motion, liquid water freezes, reflected by weakly negative $\Delta\delta^2 H_{ice}$ values near –25 ‰. As already explained in Sect. 4.1, this signal stems from the lower equilibrium fractionation factor

of condensation than that of vapour deposition. This analysis demonstrates that the new variable introduced in this study, disequilibrium in ice ($\Delta\delta^2 H_{ice}$), is a particularly useful measure to investigate the processes related to the formation and decay of ice clouds. This inference is supported by a theoretical perspective on this variable in Appendix A.

## 5 Convective and cirrus clouds

### 5.1 Ice cloud classification method

Section 4 already described different types of ice clouds and their associated isotope signatures. To objectively classify regions with different ice clouds, we use total column ice, also referred to as ice water path (IWP) in other studies. IWP is here computed by the sum of total column cloud ice (TQI) and total column snow (TQS). Figure 10 displays the vertical distributions of ice content, $\delta^2 H$ in ice, and disequilibrium in ice in 100 percentile bins of IWP ranked from largest amounts on the left to the lowest amounts on the right, following Gasparini et al. (2019), their Fig. 1. We analyse here data in the same region as in

Sect. 4.2 for July 2016. On the left, for high IWP, we clearly recognize the signatures of deep convection with very large ice content up to the tropopause cold point, transitioning into a cirrus shield with lower ice content towards lower IWP percentiles, and ending with an ice cloud free region on the right (Fig. 10a). The isotope signals in ice correspond to these suggested cloud types. Deep convective clouds are characterized by very enriched ice and positive disequilibrium in ice in the ice phase layer ($T$ < –38°C; Fig. 10b,c, label "1"). Cirrus clouds show, on average, near zero $\Delta\delta^2 H_{ice}$ values (Fig. 10c, label "2"). Underneath

the cirrus shield we observe small negative $\Delta\delta^2 H_{ice}$ values near –25 ‰ within the ice-phase layer ($T$ < –38°C), reflecting the sedimentation and sublimation of ice particles (Fig. 10c, label "3"). Negative disequilibrium in ice in the mixed-phase cloud layer between the 80$^{th}$-100$^{th}$ percentile range reflects a combination of (3) sedimentation and sublimation of ice, (4) downward transport of ice in convective downdrafts, and (5) a very minor influence of freezing liquid water in convective updrafts (Fig.



10c, labels "3-5"). The distribution of ice in Fig. 10 can also be interpreted as "liquid origin" ice from freezing and deposition in convective updrafts on the left and "in situ" formed ice under the influence of gravity wave activity or uplift above convective cells further to the right (Krämer et al., 2016).

Following Sokol and Hartmann (2020), Nugent et al. (2021), and Turbeville et al. (2021), we use IWP to distinguish three different categories of ice clouds:

1) Deep convection: IWP > 1 kg m$^{-2}$;
2) Convective outflow, convective remnants, and thick cirrus: 0.01 < IWP < 1 kg m$^{-2}$;
3) Thin cirrus: 0.0001 < IWP < 0.01 kg m$^{-2}$.

These three categories are referred to as "convective", "mixed", and "cirrus" clouds hereafter. In addition to these three ice cloud categories, we consider regions with IWP < 0.0001 kg m$^{-2}$ as "ice cloud free". Figure 11 shows two examples at different time instances to demonstrate the functionality of using IWP as proxy for different ice cloud categories. Deep convective regions have very large ice content (and liquid water content) that reaches from the middle into the upper troposphere and are classified as "*convective*" (see green bars in Fig. 11c,d). The regions labelled as "*mixed*" clouds manifest as a combination of convective outflow, remnants of convective storms, and thick cirrus. Extensive regions with relatively low ice content in the upper troposphere are classified as "*cirrus*". Some of these cirrus regions, approximately 30%, have larger TQI than TQS (Fig. 10d). This subcategory is used in a later analysis to separate atmospheric columns with predominantly suspended ice crystals from those where precipitating snow dominates IWP. Cirrus regions with TQI > TQS are marked by black hatching in the blue cirrus labels in Fig. 11c,d.

## 5.2 Isotope signals in convective and cirrus clouds

Figure 12 shows vertical profiles of the statistical distributions and averages of ice content, $\delta^2$H in ice, and disequilibrium in ice for the three ice cloud categories. The data in Fig. 12 only includes grid points with ice content $\geq 10^{-4}$ mg kg$^{-1}$, while average values are based on all grid points in the corresponding cloud regions. The three ice cloud regimes convective, mixed, and cirrus clouds correspond to 3.8 %, 28.7 %, 45.2 %, of the grid points respectively, while the remaining 22.3 % belongs to ice cloud free regions. By definition, deep convection is characterized by very large amounts of ice content (~1 g kg$^{-1}$) throughout the middle and upper troposphere (Fig. 12a). Around the cold point tropopause near 100 hPa, the average ice content is far outside the 10-90$^{th}$ percentile range, reflecting rare overshooting convection that brings very large amounts of ice at these high altitudes. Cirrus clouds display relatively low ice content in the upper troposphere, peaking around ~200 hPa with average values near 2 mg kg$^{-1}$, whereas "mixed" clouds exhibit moderate ice content near 25 mg kg$^{-1}$ throughout the middle and upper troposphere.

Average $\delta^2$H in ice within deep convection demonstrate roughly constant $\delta$ values with height near –140 ‰ (Fig. 12b). In the upper troposphere, ice in deep convection is very enriched as compared to ice in mixed and cirrus clouds (Fig. 12b). For





example, at 200 hPa the average $\delta^2$H in ice is –151 ‰ in deep convective clouds as compared to –321 ‰ and –390 ‰ for mixed and cirrus clouds, respectively. In the lower troposphere, this pattern reverses somewhat, and ice in deep convective

regions is on average slightly more depleted than ice in mixed and cirrus clouds. This pattern is very similar to the vertical profiles of $\delta^2$H in snow versus cloud ice (Fig. 7b, red and blue lines), and shows that strong convective transport of the frozen condensate dominates the isotope signature in ice in deep convective regions. The transition of very enriched ice in the upper troposphere to slightly depleted ice in the lower troposphere in deep convective clouds as compared to mixed and cirrus clouds also clearly emerges in vertical profiles of disequilibrium in ice (Fig. 12c). Average $\Delta\delta^2$H$_{ice}$ values in deep convection range

from over +300 ‰ in the upper troposphere to almost –90 ‰ in the lower troposphere. In contrast, average $\Delta\delta^2$H$_{ice}$ values in cirrus are close to zero throughout the troposphere, consistent with in situ formed ice under equilibrium fractionation, in agreement with earlier findings in Sect. 4.

Figure 13 shows the vertical profiles of the statistical distributions and averages of total water content, $\delta^2$H in total water, and

$\delta^2$H in vapour for the three ice cloud categories as well ice cloud free regions. Total water is here defined by the sum of vapour, cloud ice, snow, cloud water and rain, hereafter referred to as water for simplicity. Deep convection contains about 1-2 orders of magnitudes more water than mixed, cirrus and ice cloud free regions (Fig. 13a). Interestingly, $\delta^2$H in water in deep convection is much more enriched in the upper troposphere ($p < 500$ hPa) and slightly more depleted in the lower troposphere ($p > 500$ hPa) than in other cloud regions (Fig. 13b). These profiles show how vertical transport by deep convection acts to

redistribute the isotopic composition of water throughout the troposphere. Average $\delta$ values in vapour are relatively similar in the three ice cloud categories and the ice cloud free regions (Fig. 13c). Closer inspection shows that average $\delta^2$H$_{vap}$ in deep convection is slightly more enriched near 300 hPa and slightly more depleted in the lower troposphere ($p > 600$ hPa) as compared to $\delta^2$H$_{vap}$ in ice cloud free regions. This effect is consistent with previous studies, which suggested that deep convection enriches vapour at the approximate level of convective outflow and depletes low tropospheric vapour as a result of

depleted upper tropospheric air entrained in convective downdrafts and rain evaporation (Bony et al., 2008; Risi et al., 2008b; Blossey et al. 2010; Randel et al., 2012). In the upper troposphere (150-300 hPa), the distribution of $\delta^2$H in vapour shows a larger spread in deep convective regions compared to mixed, cirrus, and ice cloud free regions. This may stem from the fact that deep convection has a much lower sample size than the other ice cloud categories, but more likely reflects the strong opposing effect of updrafts and downdrafts on the vapour isotopic composition.


To assess the impact of vertical motion on the isotopic composition of vapour in different cloud regimes, we construct the frequency distribution of $\delta^2$H in vapour overlaid with the average vertical motion for regions with deep convection and cirrus (Fig. 14). Note that the cirrus clouds in this analysis only contain the cirrus regions where TQI > TQS (12.3% of all grid points) to remove atmospheric columns where snow particles dominate suspended ice crystals. In the upper troposphere, the range in

$\delta^2$H values in vapour within deep convection is much larger than that in cirrus regions and reaches from about –800 to



–350 ‰ (Fig. 14a). Importantly, very depleted vapour is characterised by strong upward motion, whereas very enriched vapour is associated with strong downward motion. For example, at 125 hPa very depleted vapour with $\delta^2$H values between –832 and –768 ‰ coincides with an average upward motion of 2.1 m s$^{-1}$, while very enriched vapour between –384 and –320 ‰ at this altitude experiences an average downward motion of –0.9 m s$^{-1}$. These findings are in line with the discussion in Sect. 4 and

are here shown to be directly related to deep convection. The observed pattern reverses in the lower troposphere at $p > 500$ hPa where relatively enriched (depleted) vapour is associated with upward (downward) motion. Likely, this effect results from the transport of enriched vapour in updrafts (which dominates the depletion of vapour by the preferential condensation and deposition of heavy isotopes at these lower altitudes) and depleted vapour in downdrafts, further depleted by rain evaporation or equilibration with falling rain drops.


Interestingly, the relationship between the isotopic composition of vapour and vertical motion in cirrus demonstrates a reversed pattern in the upper troposphere as compared to deep convection. Relatively depleted vapour in cirrus is associated with weak downward motion (~ –0.02 m s$^{-1}$) between 125-250 hPa, whereas enriched vapour goes along with weak upward motion (~ 0.04 m s$^{-1}$) between 175-350 hPa (Fig. 14b). Most likely, these signatures reflect the transport of depleted vapour from higher

altitudes by large-scale subsidence and enriched vapour from lower altitudes by large-scale upwelling. Thus, isotope signals in vapour differ remarkably between regions dominated by convective activity and the large-scale circulation, respectively. Fractionation during phase changes dominate the isotopic signals in vapour in regions of deep moist convection, while the direct vertical transport of vapour is of first order importance in regions dominated by the large-scale circulation.

**5.3 Impact of deep convection on the TTL water budget**

In this last section of the results, we assess the impact of deep convection on the water budget of the TTL. Figure 15 shows the partitioning of the total water and HDO budgets across the four ice cloud regimes. Opaque colours correspond to ice and the transparent colours to water in the vapour and liquid phases. In the lower troposphere ($p > 500$ hPa), the total water is proportionally distributed across the different ice cloud categories. In the upper troposphere and the lower TTL (125-200 hPa), deep convection contributes to more than 40 % to the total water budget (Fig. 15a). This points to a disproportionally large

influence of deep convection on the TTL water budget, considering that convective regions only constitute about 3.8% of the atmosphere. Contributions of deep convection to heavy deuterated water HDO are even larger and reach ~ 60 % at these high altitudes (Fig. 15b). Almost all water and deuterated water in deep convection in this part of the atmosphere consists of ice (opaque colours in Fig. 15a,b). Note that ice fractions in the mixed and cirrus clouds are much lower as compared to deep convective regions where vapour contributes most to the total water amounts. These findings underscore the key role of deep

convection in modulating the water budget of the lower TTL and its isotopic composition through convective ice lofting.





## 6 Conclusions and outlook

This study investigated the atmospheric processes related to tropical ice clouds and deep convection using regional convection-permitting model simulations in the WAM for June-July 2016. The motivation of this work is to use stable water isotope

tracers to better understand these processes, and how they affect the water budget of the TTL. This work adds three important aspects to previous studies: (1) whereas previous stable water isotope modelling studies used conceptual frameworks, Lagrangian trajectory calculations, single column models, large eddy simulations, idealized two-dimensional cloud-resolving models, and global circulation models (Dessler and Sherwood, 2003; Dessler et al., 2007; Bony et al., 2008; Blossey et al., 2010; Sayres et al., 2010; Eichinger et al., 2015), we use for the first time regional convection-permitting model simulations

constrained by actual meteorological conditions, (2) building upon observational and modelling insights into the formation and decay of tropical ice clouds (Krämer et al., 2016; Urbanek et al., 2017; Gasparini et al., 2019, 2021), we present an integrated view on these processes based on stable water isotope information, and (3) we contribute to the longstanding debate on processes that affect the TTL water budget and show, consistent with previous observations and modelling efforts (Moyer et al., 1996; Kuang et al., 2003; Steinwagner et al., 2010; Bolot and Fueglistaler, 2021), that deep convection strongly affects

the TTL water budget through convective ice lofting.

First, we evaluated the credibility of our regional model simulations by comparing model output from simulations with different resolutions and treatments of convection to observations of isotopes in monthly precipitation. Our COSMO$_{iso}$ simulations reasonably represent the spatial distribution of monthly precipitation isotopes as compared to GNIP observations.

Calculation of root mean square errors showed that the convection-permitting simulation with a horizontal grid spacing of 7 km and 60 vertical model levels outperforms the simulations with parameterized and explicit convection at a horizontal grid spacing of 14 km. For the remaining analysis of tropical ice clouds, we used the convection-permitting simulation with a 7 km horizontal grid spacing, which also represents very well the different stages in the monsoon evolution as observed during the DACCIWA field campaign in 2016 (Knippertz et al., 2017). Also, this simulation exhibits realistic behaviour of MCSs that

move westwards over North Africa during the course of several days.

A case study of an MCS and complementary statistical analysis reveal how tropical deep convection modulates the isotopic composition of water vapour and ice. Five key processes could be identified that are related to the formation and decay of tropical ice clouds as schematically depicted in Fig. 16:

1)   Convective updrafts transport enriched ice into the upper troposphere going along with a strong depletion of the vapour by the preferential condensation and deposition of heavy isotopes in these updrafts.

      2)   In contrast, ice in widespread cirrus shields is in approximate isotopic equilibrium with the environmental vapour, suggesting in situ ice formation under equilibrium fractionation.



3) Sedimentation and sublimation of ice is evident in the mixed-phase cloud layer in convective regions and underneath
cirrus shields as derived from moderately negative disequilibrium in ice.

4) Convective downdrafts go along with non-fractionating sublimation of ice that enriches the environmental vapour.

5) At the base of convective updrafts, freezing of liquid water occurs, reflected by weakly negative disequilibrium in ice as
    a result of a lower equilibrium fractionation factor of condensation than that of vapour deposition.

This study also introduces a new variable, disequilibrium in ice (see Appendix A for more information), that appears very
useful to identify these five key processes.

Particularly striking is the large variability in the isotopic composition of water vapour in the upper troposphere and lower
TTL (100-200 hPa). $\delta^2$H values can reach as low as –800 ‰ in strong convective updrafts and up to –400 ‰ in strong
convective downdrafts. Previous studies already noted a very large range of $\delta^2$H in vapour based on observations at these
altitudes (Webster and Heymsfield, 2003; Hanisco et al., 2007; Sayres et al., 2010). We show that moist processes in convective
updrafts and downdrafts are responsible for these large fluctuations, namely vapour depletion due to condensation and
deposition in updrafts and vapour enrichment from non-fractionating sublimation of ice in downdrafts. These processes
strongly affect the environmental vapour since approximately 50% of total water consists of ice at these altitudes (see Figs. 7b
and 15). In contrast, water vapour in cirrus regions that are dominated by suspended ice crystals shows a reversed relationship
between its isotopic composition and vertical motion: Enriched water vapour is associated with (weak) ascending motion and
depleted vapour with (weak) descending motion, suggesting a first order importance of large-scale vertical transport of water
vapour. Thus, isotope signals in water vapour can hold important clues to assess whether the atmospheric transport is
dominated by convection or the large-scale circulation.

Importantly, our model simulations show that tropical deep convection is not only of key relevance for the formation of ice
clouds, but also strongly modulates the water budget of the lower TTL (100-150 hPa). Deep convection, although only
occurring at about 3.8 % in time and space, contributes to about 40% of the total water budget and even up to 60 % of the
deuterated water (HDO) budget in this part of the atmosphere. By large, most of the water in deep convection consists of ice,
underlining the key relevance of convective ice lofting for the water budget of the lower TTL. Recently, Bolot and Fueglistaler
(2021) showed that most ice entering this region through convection immediately falls out. An open question in this context is
to what extent both depleted and enriched local vapour signatures within regions of deep convection can affect the water
vapour budget in the entire TTL. A follow-up study could address this question by budget calculations and forward trajectories
from convective outflows to assess the spatiotemporal scales across which these isotope signatures are transported in the TTL.

Before concluding, we list a few limitations of this work. First, the model simulations used a relatively simplistic one-moment
microphysics scheme with two ice species only (suspended ice crystals and sedimenting snow particles), with inherent
limitations in the representation of complex cloud microphysical processes. Second, our analysis of tropical ice clouds is based





on model simulations with a horizontal grid spacing of 7 km and 60 model levels. Although this resolution is considered appropriate for a convection-permitting setup, and the simulated MCSs demonstrate a realistic behaviour, the results of this study may be sensitive to the chosen resolution. Third, our study focuses on the WAM region during the period of June-July 2016. Therefore, it is not known to what extent our results are representative for other (monsoon) regions and for other periods. For instance, it is conceivable that deep convection over the Maritime Continent and the Amazon Basin differs in certain aspects from the WAM-related convection investigated in this study.

Overall, this study demonstrates the usefulness of stable water isotopes as a natural tracer of the Earth's water cycle. Based on the two-month simulations for the WAM in 2016, we show that stable water isotopes can reflect different processes related to the formation and decay of tropical ice clouds and that tropical deep convection strongly affects the water budget of the lower TTL through convective ice lofting. These findings can be complemented in future work by using (i) a quasi-global and year-round analysis, (ii) higher resolution simulations with more advanced ice cloud microphysics schemes, (iii) online trajectories in high-resolution simulations to trace the moist processes and their isotopic signatures along liquid-origin and in situ ice cloud formation pathways, and (iv) budget calculations and forward trajectories to quantify the remote impact of local deep convection on the TTL water vapour.



**Appendix A. Disequilibrium in ice**

The isotopic disequilibrium in ice with respect to local vapour conditions is a simple measure to quantify how "exotic" the
frozen hydrometeors are in a given upper tropospheric environment (for the reader's convenience the definition from Eq. 1 in
the main text is repeated here):

$$\Delta\delta^2H_{ice} = \delta^2H_{ice} - \delta^2H_{ice,eq}, \text{ (Eq. A1)}$$

where $\delta^2H_{ice,eq}$ is obtained as a function of temperature from the local vapour phase $\delta^2H_{vap}$ using the Merlivat and Nief
(1967) equilibrium fractionation factor also used in the COSMO$_{iso}$ model for ice formation from vapour deposition. The $\delta^2H_{ice}$
is obtained as a mass weighted mean from the $\delta^2H$ of snow and cloud ice water contents.

Here we shortly summarise the expected range of numerical values of $\Delta\delta^2H_{ice}$ for the five key processes discussed in Sect. 4.
$\Delta\delta^2H_{ice}$ reflects (i) vertical displacements due to transport or sedimentation of frozen hydrometeors, which were formed (at
least partly) in an isotopically different environment, and (ii) phase change processes such as water vapour deposition,
sublimation of ice or freezing liquid. Depending on the environmental conditions (updrafts, downdrafts, no vertical motion)
either vertical displacements, or phase changes, or a combination of the two are relevant.

(1) **Convective lofting of enriched ice into the upper troposphere**:
The upper parts of deep convective updrafts are always associated with large positive $\Delta\delta^2H$ values (see Figs. 5h, 6c, 8c
680       and 9e). This reflects the fact that the bulk of the ice with isotope ratio $R_{ice}$ found in these strongly ascending
environments has formed at lower altitudes:

$$R_{ice} > \alpha_{eq}R_{vap},$$

with $\alpha_{eq}(T) = \frac{R_{ice}}{R_{vap}} > 1$ and assuming $\frac{\partial R_{vap}}{\partial p} > 0$, where $R_{vap}$ is the isotope ratio of the local vapour. When treating the
hydrometeor categories ice and snow separately (Fig. 7b), it becomes clear that the small ice crystals (cloud ice) that
685       most likely formed recently in the updraft are on average in approximate equilibrium with the local vapour $R_{cloud\ ice} = \alpha_{eq}R_{vap}$ (see blue line for $\delta^2H_{cloud\ ice}$ vs. turquoise line for $\delta^2H_{ice,eq}$ in Fig. 7b). The larger snow hydrometeors,
however, are much more enriched than the ice that would form from local vapour at altitudes with $p < 400$ hPa (red line
for $\delta^2H_{snow}$ vs. turquoise line for $\delta^2H_{ice,eq}$ in Fig. 7b), indicating that the bulk of this snow is transported upward by
strong ascending motion from lower altitudes. The $\Delta\delta^2H_{ice}$ induced by convective lofting can amount to a maximum of
690       400 ‰, when considering the range of $\delta^2H_{ice,eq}$ values in the troposphere (i.e. ice formed at 600 hPa that is transported
up to 150 hPa). Local variability, however, can lead to much larger positive $\Delta\delta^2H_{ice}$ values, as shown in Figs. 8c and
9e with values > 600 ‰ between 100-150 hPa. This can be explained by the excessive depletion of heavy isotopes in
very strong convective updrafts, leading to isotopic compositions of vapour below -750 ‰ in the upper core regions of
the updrafts, and $\Delta\delta^2H_{ice}$ values well above 400 ‰ (Figs. 8c and 9d,e).




(2) **In situ ice formation under equilibrium fractionation**:
In this case $R_{\text{ice}} = \alpha_{\text{eq}} R_{\text{vap}}$, hence, $\Delta\delta^2\text{H}_{\text{ice}} = 0$, with slight variations of $\pm 10$‰ (see Fig. 5h, 6c and 8c), which may

be due to turbulent entrainment or detrainment of ice in the upper tropospheric environment, or due to non-equilibrium

fractionation in super-saturated environments.


(3) **Sedimentation and sublimation of ice in the mixed-phase cloud layer of convective systems and the lower parts of cirrus shields:**
Sedimentation leads to a slow descent of the large ice particles. If they formed at higher altitudes, e.g. from aggregation

and water vapour deposition on in situ formed ice, then

$$R_{\text{ice}} < \alpha_{\text{eq}} R_{\text{vap}},$$

again assuming $\frac{\partial R_{\text{vap}}}{\partial p} > 0$, which leads to $\Delta\delta^2\text{H}_{\text{ice}} < 0$. Sedimentation in the upper troposphere can in principle lead to

$\Delta\delta^2\text{H}_{\text{ice}}$ values as low as –400‰ if snow formed at the level of the $\delta^2\text{H}_{\text{vap}}$ minimum at about 150 hPa and sedimented

down to the melting level at about 600 hPa following the same rationale as for process (1), see Fig. 7b. However,

minimum values of about –200‰ are typically observed in upper tropospheric environments with weak vertical winds

below cirrus shields (at about 200 hPa) or just above the melting level (see Fig. 5h and 6c). This is consistent with the

observation that snow formed in the upper troposphere from in situ ice usually sublimates completely in the upper

troposphere. The effect of sublimation on $\Delta\delta^2\text{H}_{\text{ice}}$ is discussed below.

(4) **Convective downdrafts, in which ice sublimates:**
The effect of sublimation cannot be directly quantified without additional information about sublimation fluxes.

Nevertheless, the maximum impact of sublimation on $\Delta\delta^2\text{H}_{\text{ice}}$ can be estimated by choosing the extreme case of ice

sublimation into a totally dry environment. Since sublimation is assumed to be non-fractionating, $R_{\text{vap}} = R_{\text{ice}}$, and thus:

$$R_{\text{ice,eq}} = \alpha_{\text{eq}} R_{\text{ice}},$$

which, when substituting into Eq. 3 and using $\delta^2\text{H}_{\text{ice}}$ instead of $R_{\text{ice}}$ gives:

$$\Delta\delta^2\text{H}_{\text{ice}} = (1 - \alpha_{\text{eq}})(\delta^2\text{H}_{\text{ice}} + 1000).$$

If we assume a typical upper tropospheric $\delta^2\text{H}_{\text{ice}}$ vertical profile, such as shown in Fig. A1, we find $\Delta\delta^2\text{H}_{\text{ice}}$ values

between –140‰ and –100‰ (green line in Fig. A1).

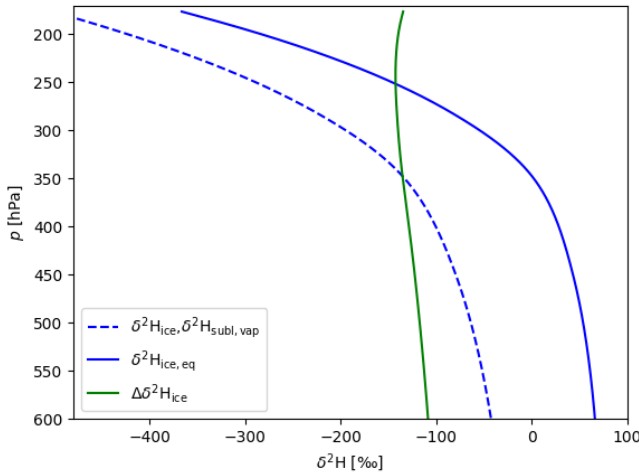

Figure A1: $\Delta\delta^2H_{ice}$ (green line) obtained from a hypothetical experiment, in which ice with the isotope composition $\delta^2H_{ice}$ is inserted into a totally dry environment, moistened by sublimation of ice only. Equilibrium ice $\delta^2H_{ice,eq}$ from this newly formed water vapour is shown by the thick blue line. The $\delta^2H_{ice}$ profile in this example follows a Rayleigh curve obtained from a moist adiabatic ascent (dashed thick blue line) with $T_0$=30°C at 1000 hPa.

Typically, the background air in the upper troposphere is not totally dry, as assumed in this idealised example and therefore the $\Delta\delta^2H_{ice}$ in an environment with sublimation will be less negative and will depend on the pre-existing specific vapour content, the sublimation flux, and $\delta^2H_{vap}$. The case study and statistical analysis typically show moderate negative $\Delta\delta^2H_{ice}$ values ~ –50 to –120 ‰ (Figs. 5h and 8c), qualitatively consistent with the extreme scenario discussed here.

(5) **The freezing of liquid water in the lower parts of convective updrafts in the mixed-phase cloud layer**:
The freezing of liquid droplets is assumed to be non-fractionating in COSMO_{iso}. Due to the stronger fractionation between vapour and ice than between vapour and liquid (Fig. A2), liquid-origin ice formed via the condensation and freezing pathway will lead to a slightly negative $\Delta\delta^2H_{ice}$ near –20‰ (Fig. A3). We expect that the impact of this process on $\Delta\delta^2H_{ice}$ is limited to the base of updrafts in mixed phase clouds and that aloft it is rapidly outbalanced by the effect of vertical transport of aggregated ice (process 1).





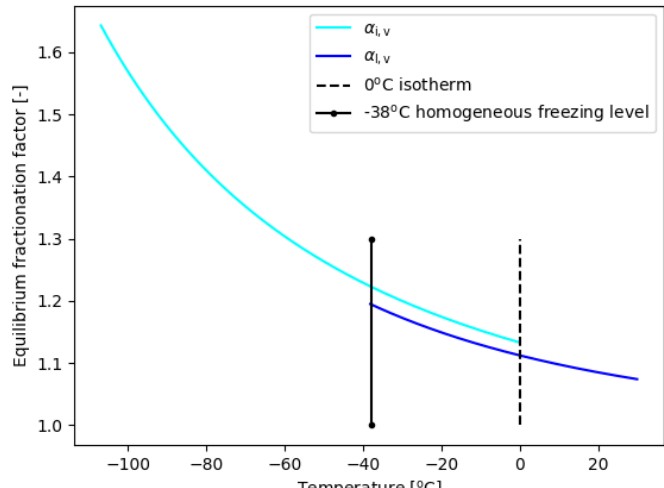

Figure A2: Difference between liquid and ice equilibrium fractionation factors. Inspired from Bolot et al. 2013 (their Fig. 1a).


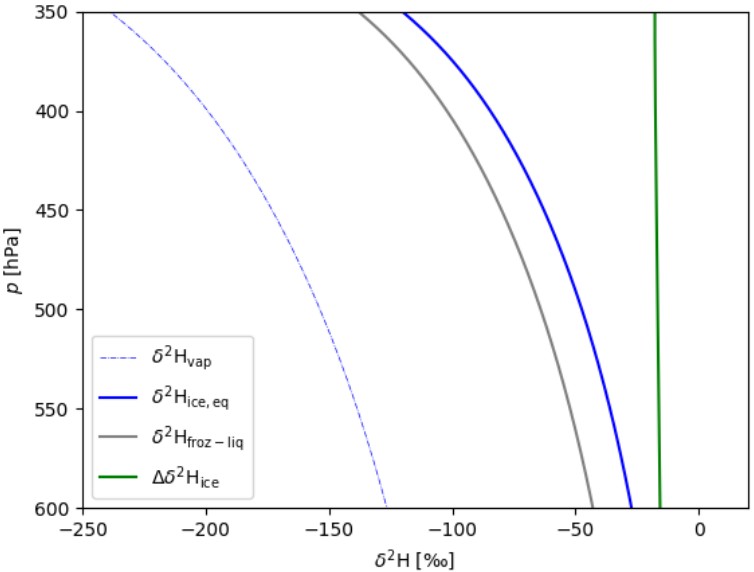

Figure A3: Expected $\Delta\delta^2 H_{ice}$ (green line) when assuming a typical moist adiabatic profile of upper tropospheric $\delta^2 H_{vap}$ (dashed thin blue line) with $T_0=30°C$ at 1000 hPa. The equilibrium ice formed from this vapour is shown by the thick blue line and the ice formed from freezing of locally condensed vapour by the thick grey line.




## Code and data availability

GNIP observations from IAEA and WMO are available under https://nucleus.iaea.org/wiser. Selected output from our COSMO$_{iso}$ simulations is available upon requests to the authors. The National Center for Atmospheric Research (NCAR) command language (NCL) version 6.5.0 has been used for most computations and visualisation of the results.

## Author contribution

All authors contributed to the conceptualisation of the study. HW acquired funding and supervised the progress of this project. AJdV conducted the research, visualized the results and wrote the paper. FA, SP and HW substantially contributed to the discussion of the results and the writing of the manuscript. FA conceptualised the variable disequilibrium in ice ($\Delta\delta^2 H_{ice}$) and wrote Appendix A.

## Acknowledgements

The authors would like to thank Fabienne Dahinden and Lukas Papritz (both ETH Zurich) for technical support, and Peter Blossey (University of Washington) and Stephan Fueglistaler (Princeton University) for valuable discussions on different aspects of this study. Also, this study benefitted from inspiring and valuable discussions in the community of the Partnerships for International Research and Education (PIRE) cirrus studies (https://www.pire-cirrus.org) and the MOTIV project
(https://www.imk-asf.kit.edu/english/Projects_2285.php). We acknowledge the contribution of Martin Werner who provided ECHAM5-wiso data. The COSMO$_{iso}$ simulations were run at the CSCS under projects sm08 and sm32. Selected output from these simulations is available upon requests to the authors.

## Financial support

AJdV acknowledges financial support from the PIRE funding scheme via the Swiss National Science Foundation (SNSF) grant
no. 177996.



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



Figures



**Figure 1.** Overview of the WAM system and the COSMO$_{iso}$ model domain. Monthly precipitation (mm) in shaded and monthly mean circulation patterns in contours from the convection-permitting COSMO$_{iso}$ simulation at 7 km horizontal grid spacing for July 2016. The circulation patterns show the Saharan heat low (mean sea level pressure, hPa, in black contours), mid-tropospheric anticyclone (500-hPa geopotential height, gpm, in orange contours), the moist air inflow from the eastern tropical Atlantic (vertically integrated horizontal water vapour transport between 1000-850 hPa in red vectors where the IVT magnitude > 50 kg m$^{-1}$ s$^{-1}$), and the African easterly jet (600-hPa wind in blue vectors where the wind velocity > 7 m s$^{-1}$). The COSMO$_{iso}$ model domain (thick black contours) has a rotated grid, is centred at 7.5°E, 13°N, and spans approximately from 36°W to 51°E and 20°S to 46°N. The box in the centre denotes the region of interest (thin black contours; 10°W-20°E, 5°-20°N) used for the statistical analysis in Sects. 4.2, 4.3, and 5.







**Figure 2.** Monthly precipitation and its isotopic composition from the three model simulations and GNIP observations for July 2016. The columns show (left) precipitation amounts (mm), (middle) $\delta^2$H in precipitation (‰), and (right) $d$-excess in precipitation (‰). The rows show the data from (a)-(c) PAR14, (d)-(f) EXPL14, (g)-(i) EXPL7, (j)-(l) GNIP observations, and (m)-(o) values of GNIP observations and COSMO$_{\text{iso}}$ simulations, as indicated by the legend, for station locations ordered from west to east. Open black circles in (j-l) denote missing values. In (m-o) COSMO$_{\text{iso}}$ output is interpolated at the station locations, and values are only plotted if the simulated precipitation amounts are larger than 1 mm month$^{-1}$.





**Figure 3.** Temporal evolution of precipitation from EXPL7 during the period of June-July 2016. In (a) domain averages of precipitation amounts (mm hour⁻¹, in black) and $\delta^2H$ in precipitation (‰, in green) over the region of interest (10°W-20°E, 5°-20°N; see the black box in Fig. 1) with hourly values in thin lines and hourly smoothed values over a 24-hour running time window in thick lines. In (b) the temporal evolution of precipitation amounts (mm hour⁻¹) as a function of latitude averaged over the longitude band of 10°W to 20°E. In (c) a Hovmöller diagram of precipitation amounts (mm hour⁻¹) averaged over the latitude band of 5°N to 20°N. The four phases in (a-c) refer to the WAM stages discussed in Sect. 3.2 and the markers in (b,c) denote the time and location of the MCS discussed in the case study of Sect. 4.1.



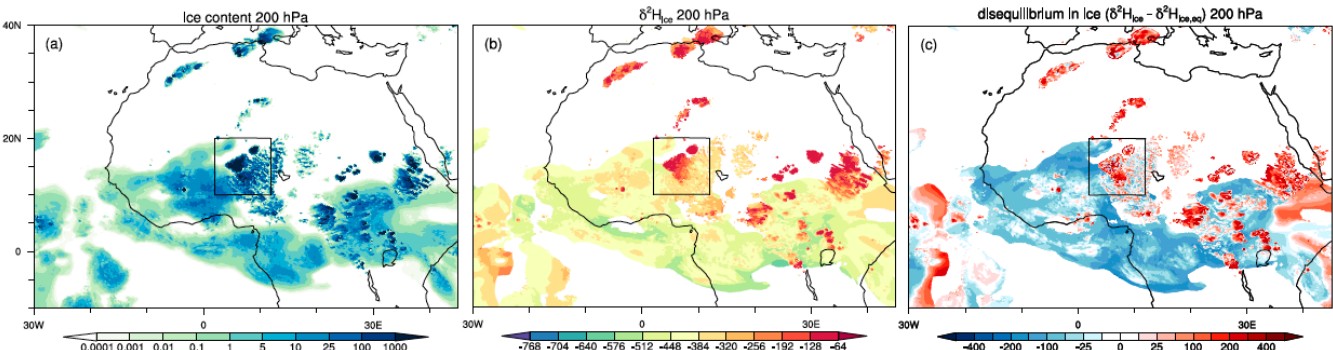

**Figure 4.** Spatial distribution of ice and its isotopic composition at 200 hPa on 15 June 2016, 18:00 UTC. In (a) ice content (mg kg$^{-1}$), (b) $\delta^2$H in ice (‰), and (c) disequilibrium in ice (‰), see the text for details. Variables in all panels are plotted transparent where ice content values are below $10^{-4}$ mg kg$^{-1}$.

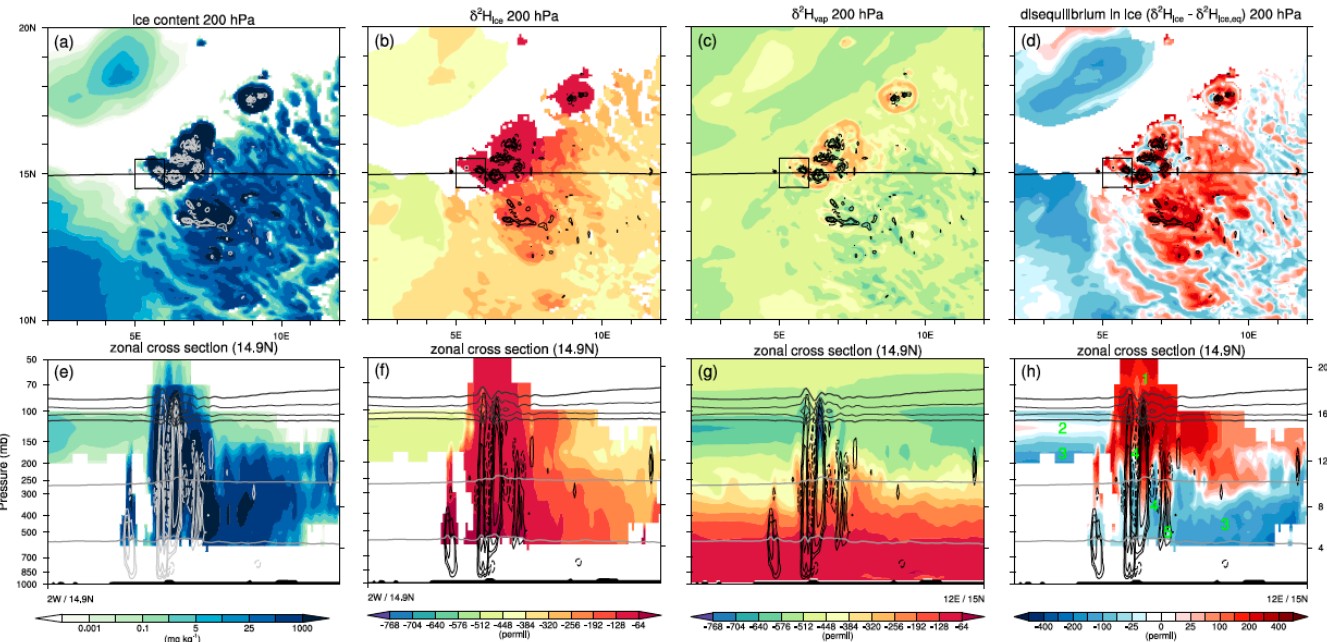

**Figure 5.** As Fig. 4, but for a smaller region (2-12°E, 10-20°N; as indicated by the black boxes in Fig. 4a-c), and at 16 UTC, 15 June 2016. In addition, (c) shows $\delta^2$H in vapour (‰), and (e-h) vertical cross sections in a zonal direction along the black line in (a-d). The light grey contours in the cross sections are the isotherms at $T = 0°C$ and $T = -38°C$ to denote the liquid, mixed-phase and ice cloud layers, and the dark grey contours are the isotherms at $T = 191$ K, 193 K, and 195 K with increasing thickness to demarcate the tropopause cold point. The very light grey contours in (a,e) and black contours in (b-d, f-h) show upward (downward) vertical motion in solid (dashed) lines at intervals of 1, 2, 5, and 10 m s$^{-1}$ in (a-d), and of 0.5, 1, 2, 5, and 10 m s$^{-1}$ in (e-h). The labels 1-5 in (h) refer to processes discussed in the text of Sect. 4.1.



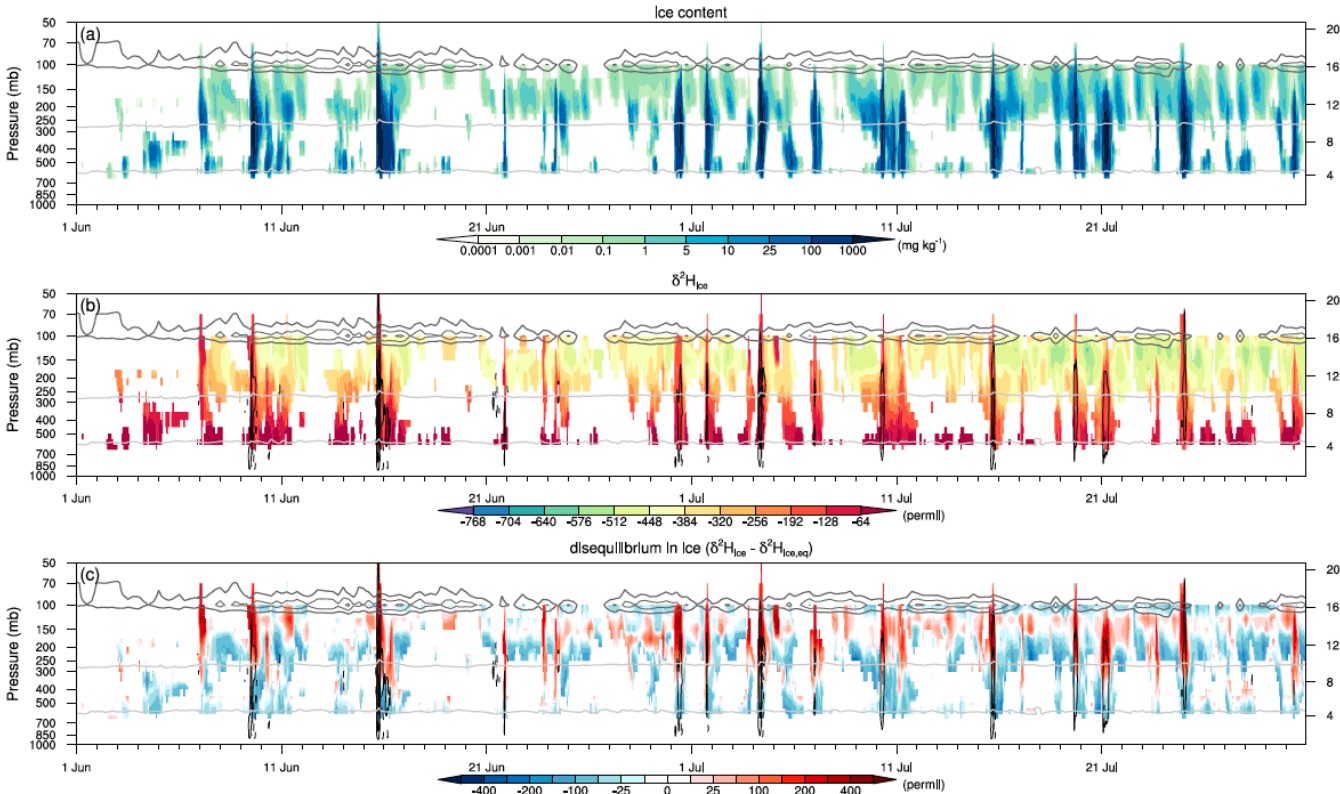


**Figure 6.** Time evolution of ice and its isotopic composition for the period of 1 June to 30 July 2016. In (a) ice content (mg kg$^{-1}$), (b) $\delta^2$H in ice (‰), and (c) disequilibrium in ice (‰), averaged over a small region (5-6°E, 14.5-15.5°N), indicated by the black boxes in Fig. 5a-d. Variables in all panels are plotted transparent where domain average ice content values are below $10^{-4}$ mg kg$^{-1}$. As in Figs. 4 and 5, the light grey contours are the isotherms at $T = 0$°C and $T = -38$°C, and the dark grey contours are the isotherms at $T = 191$ K, 193 K, and 195 K with

increasing thickness. Black solid (dashed) contours show again the upward (downward) vertical motion, but at positive (negative) intervals of 0.1, 0.5, 1, 2, and 5 m s$^{-1}$.




**Figure 7.** Vertical profiles and probability density functions water species and their isotopic composition in the target region (10°W-20°E, 5°-20°N; see black box in Fig. 1) for July 2016. Vertical profiles of domain average values in (a) of temperature (grey, K), cloud ice (blue, mg kg$^{-1}$), snow (red, mg kg$^{-1}$), and the fraction of summed cloud ice and snow of total water (green, %), and in (b) of $\delta^2$H (‰) in vapour (black), equilibrium ice (turquoise), cloud ice (blue), and snow (red) as indicated by the legend. The black dashed line in (b) is $\delta^2$H (‰) in vapour as predicted by Rayleigh distillation for moist adiabatic ascent of a saturated air parcel at 1000 hPa with a temperature of 25°C. In (c-e) the mass-based probability density functions of $\delta^2$H in water species as in (b) at 125, 200 and 450 hPa, complemented by the frequency-based probability density functions of snow (red dashed). The $\delta^2$H in cloud ice and snow in (b-e) is based on grid points with cloud ice and snow content $\geq 10^{-4}$ mg kg$^{-1}$.

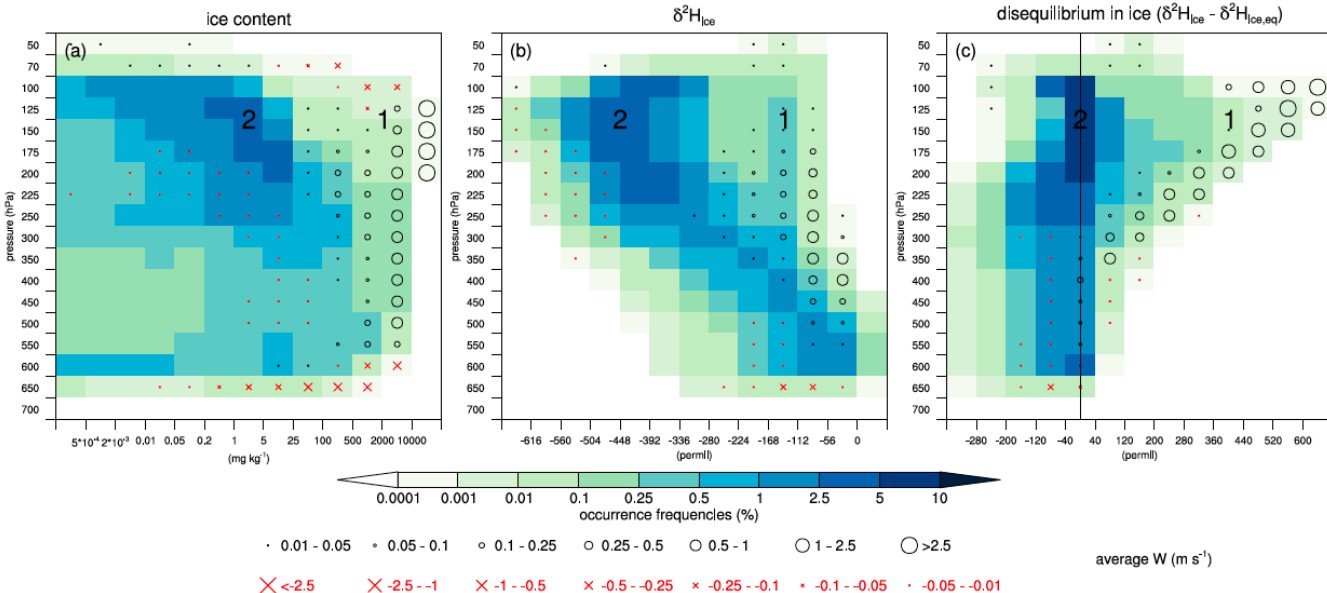

**Figure 8.** Frequency distribution of ice and its isotopic composition across pressure levels in the region of interest for July 2016. Frequency occurrences of (a) ice content (mg kg$^{-1}$), (b) $\delta^2$H in ice (‰), and (c) disequilibrium in ice (‰) are based on grid points with ice content $\geq$ 10$^{-4}$ mg kg$^{-1}$. Black circles (red crosses) denote the positive (negative) average vertical motion for the corresponding bins, as indicated by the legend, and are only plotted for bins where the frequency occurrences exceed 10$^{-4}$ %. The labels (1-2) in the panels refer to the processes discussed in the text of Sect. 4.2.



**Figure 9.** Frequency distribution and isotopic composition of water species as a function of vertical motion across pressure levels in the region of interest for July 2016. In (a) frequency occurrences (%), (b) ice content (mg kg$^{-1}$), (c) $\delta^2$H in ice (‰), (d) $\delta^2$H in vapour (‰), (e) disequilibrium in ice (‰), and (f) relative humidity with respect to ice (%). Values in (b-f) are plotted transparent where frequency occurrences in (a) are below $10^{-5}$ % and in (c,e) also if average ice content falls below $10^{-4}$ mg kg$^{-1}$. Note the irregular intervals and asymmetric distribution of the vertical motion bins as indicated on the x-axes.





**Figure 10.** Vertical distribution of ice and its isotopic composition as a function of IWP in the region of interest for July 2016. Vertical profiles of (a) ice content (mg kg$^{-1}$), (b) $\delta^2$H in ice (‰), (c) disequilibrium in ice (‰), and (d) IWP amounts (blue; kg m$^{-2}$) and the fraction of grid points with TQI > TQS (%), are constructed using 100 percentile bins ranked on IWP. $\delta^2$H in ice and disequilibrium in ice are plotted transparent where average ice content is below 10$^{-4}$ mg kg$^{-1}$. As in Figs. 5 and 6, the light grey contours are the isotherms at $T$ = 0°C and $T$ = –38°C, and the dark grey contours are the isotherms at $T$ = 195 K.





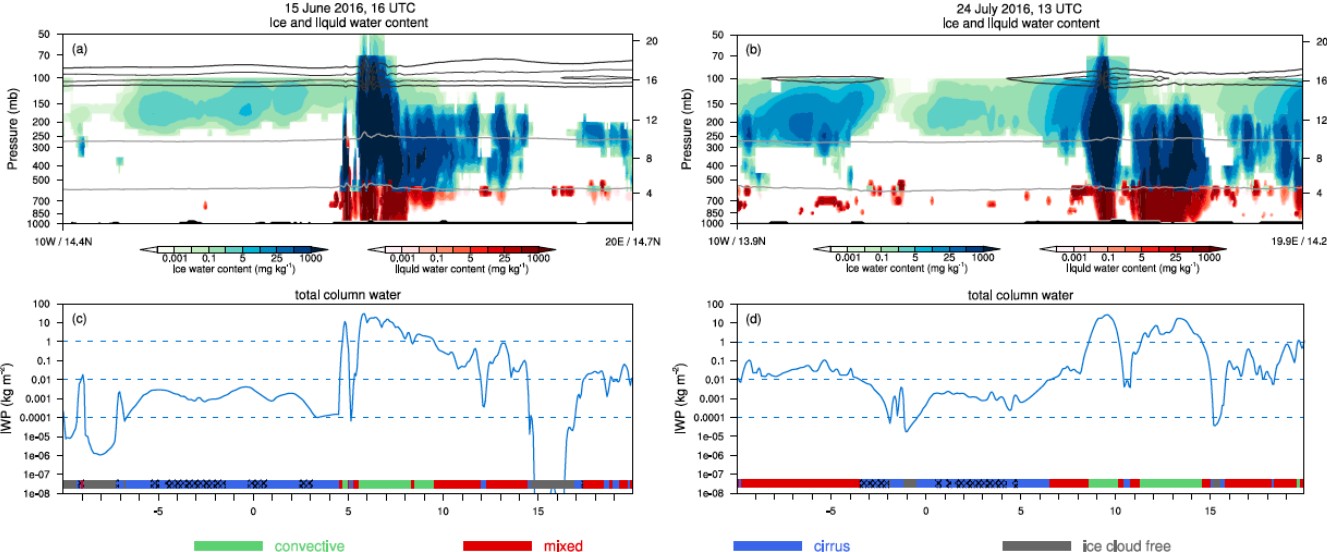

**Figure 11.** Illustrative examples of the different categories of ice clouds. In (a,b) vertical cross sections in a zonal direction with ice water content in blue (mg kg$^{-1}$) and liquid water content in red (mg kg$^{-1}$), and (c,d) the corresponding IWP amounts (blue, kg m$^{-2}$) at (a,c) 15 June 2016, 16 UTC, and (b,d) 24 July 2016, 13 UTC. The horizontal lines in (c,d) indicate the thresholds applied on IWP (light blue) to distinguish the different ice cloud types. The coloured bars in the lower parts of (c,d) indicate the ice clouds regions as indicated by the legend below, whereby the black hatching of the blue bars indicate cirrus regions where TQI > TQS, see the text for details.



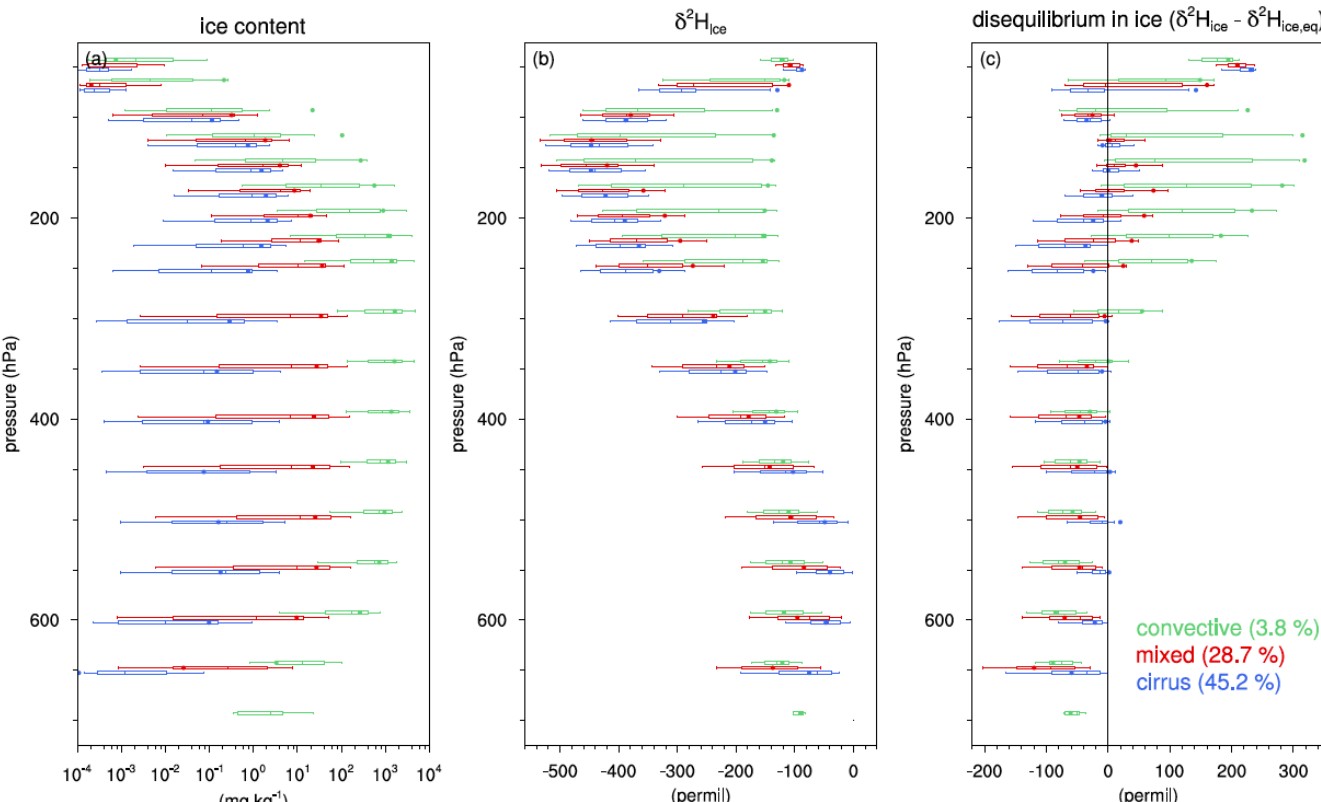


**Figure 12.** Statistical dispersion and averages of ice and its isotopic composition for the pressure range 50-700 hPa in the region of interest in July 2016. Boxes and whiskers indicate the 10th, 25th, median, 75th, and 90th percentiles, and the dots the averages, of (a) ice content (mg kg$^{-1}$), (b) $\delta^2$H in ice (‰), and (c) disequilibrium in ice (‰) for the ice cloud categories as indicated by the legend. The statistical dispersion is based on grid points with ice content $\geq 10^{-4}$ mg kg$^{-1}$, while the averages are based on all grid points within the respective cloud type regions.




**Figure 13.** As Fig. 12, but for (a) total water content (g kg$^{-1}$), (b) $\delta^2$H in total water (‰), and (c) $\delta^2$H in vapour (‰) for the pressure range 50-1000 hPa. In addition, the information for ice cloud free regions is included in black.






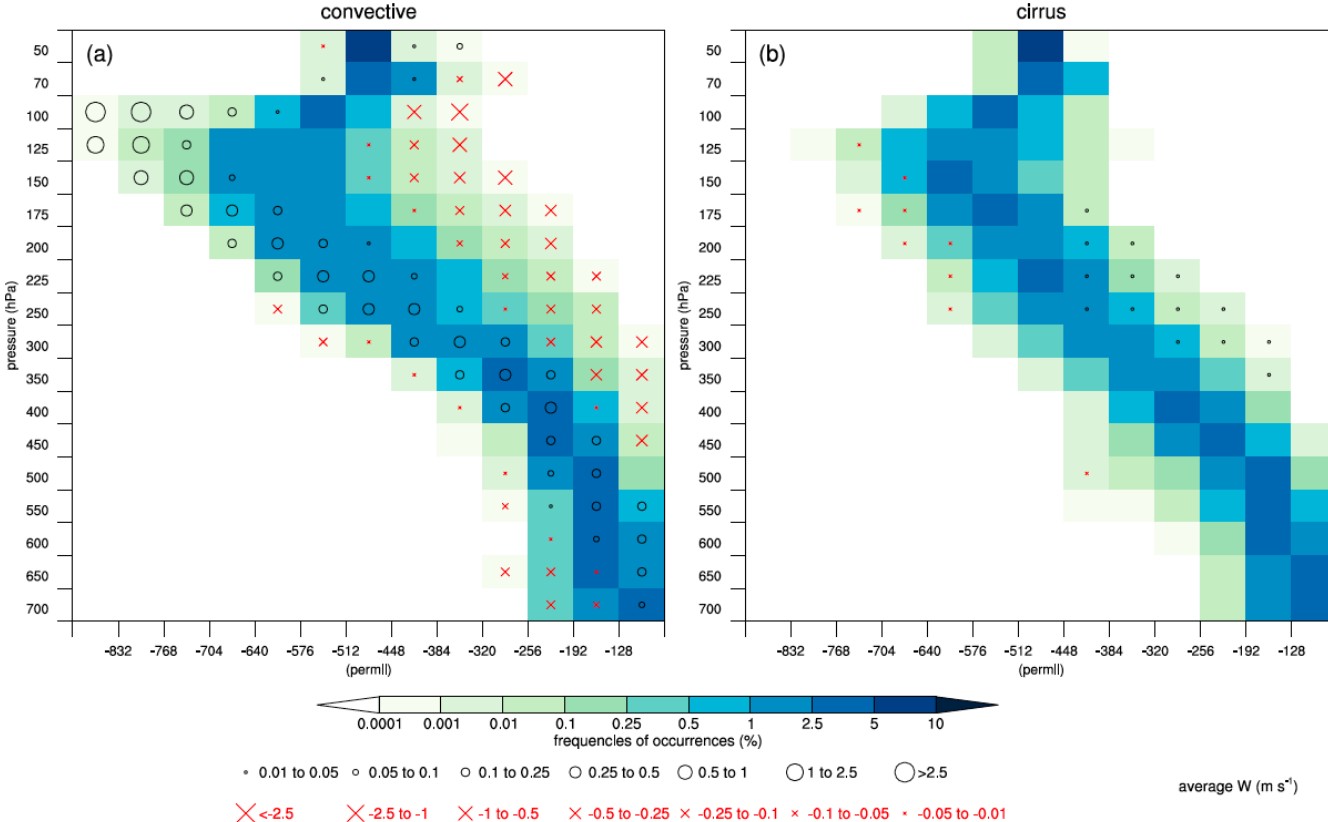

**Figure 14.** Frequency distribution of the isotopic composition of vapour and average vertical motion in the region of interest in July 2016. Frequency occurrences (%) of $\delta^2$H in vapour for the ice cloud categories (a) deep convection and (b) cirrus with TQI > TQS. Black circles (red crosses) denote the positive (negative) average vertical velocity (m s$^{-1}$) for the corresponding bins, as indicated by the legend, and are only plotted if frequency occurrences for the respective bins $\geq$ 0.0001 %.



**Figure 15.** Total water budgets partitioned over the four ice cloud categories in the region of interest in July 2016. The colours show the contributions from the four different ice cloud categories, as indicated by the legend, to (a) the total water and (b) the total deuterated water (HDO) budgets. Opaque colours represent water in the ice phase and transparent colours water in the vapour and liquid phases. The numbers in the legend indicate the fractions of grid points that correspond to each of the different ice cloud categories.



**Figure 16.** Schematic representation of the five key processes related to the formation and decay of tropical ice clouds: (1) convective lofting of very enriched ice along with the isotopic depletion of vapour through the preferential condensation and deposition of heavy isotopes in the updrafts, (2) in situ ice formation under equilibrium fractionation in cirrus shields, (3) sedimentation and sublimation of ice in the mixed-phase cloud layer of convective systems and underneath cirrus shields, (4) non-fractionating sublimation of ice in convective downdrafts that enriches the environmental vapour, and (5) freezing of liquid water in the lower parts of convective updrafts. These processes are reflected by stable water isotope signals in ice, vapour, and especially the disequilibrium in ice ($\Delta\delta^2 H_{ice}$), as indicated by the legend. See the text for details.


**Tables**

**Table 1**. Overview of the three model simulations and their configurations.

|  | horizontal grid spacing | zonal and meridional grid points | vertical levels | time step (s) | treatment convection |
|---|---|---|---|---|---|
| PAR14 | 14 | 696 x 528 | 40 | 60 | parameterized |
| EXPL14 | 14 | 696 x 528 | 40 | 60 | explicit |
| EXPL7 | 7 | 1392 x 1056 | 60 | 40 | explicit |

**Table 2.** RMSE of precipitation in the three COSMO$_{iso}$ experiments as compared to GNIP observations[a,b]

| June 2016 | precipitation (mm) | $\delta^{18}$O (‰) | $\delta^2$H (‰) | $d$ (‰) |
|---|---|---|---|---|
| nr. obs[3] | 8 | 13 | 13 |  |
| PAR14 | **95.2** | 4.30 | 31.2 | 5.61 |
| EXPL14 | 99.8 | 3.43 | 23.0 | 5.97 |
| EXPL7 | 98.0 | **2.75** | **20.4** | **4.01** |
| *July 2016* |  |  |  |  |
| nr. obs[c] | 11 | 14 | 14 |  |
| PAR14 | 138.9 | 3.13 | 23.9 | 6.81 |
| EXPL14 | 111.1 | 3.43 | 22.7 | 8.59 |
| EXPL7 | **109.5** | **2.02** | **17.7** | **5.70** |

[a] RMSE values are based on station locations where all three simulations have precipitation amounts > 20 mm month[-1].
[b] best skill scores across three model simulations are in bold.
[c] refers to the number of station locations with valid data, that its, non-missing values in GNIP and precipitation amounts > 20 mm month[-1] in all three model simulations.

**Table 3.** ME of precipitation in the three COSMO$_{iso}$ experiments as compared to GNIP observations[a,b]

| June 2016 | precipitation (mm) | $\delta^{18}$O (‰) | $\delta^2$H (‰) | $d$ (‰) |
|---|---|---|---|---|
| nr. obs[3] | 8 | 13 | 13 |  |
| PAR14 | **6.02** | −1.74 | −14.65 | **−0.69** |
| EXPL14 | 41.8 | 0.26 | **−0.22** | −2.27 |
| EXPL7 | 29.9 | **−0.10** | −3.14 | −2.33 |
| *July 2016* |  |  |  |  |
| nr. obs[c] | 9 | 12 | 12 |  |
| PAR14 | 13.7 | −1.47 | −8.76 | 3.04 |
| EXPL14 | −29.6 | 0.61 | 3.95 | **−0.94** |
| EXPL7 | **6.02** | **−0.07** | **0.66** | 1.23 |

[a] ME values are based on station locations where all three simulations have precipitation amounts > 20 mm month[-1].
[b] best skill scores across three model simulations are in bold.
[c] refers to the number of station locations with valid data, that its, non-missing values in GNIP and precipitation amounts > 20 mm month[-1] in all three model simulations.