# Peer review of "Stable water isotope signals in tropical ice clouds in the West African monsoon simulated with a regional convection-permitting model"

_Atmospheric Chemistry and Physics, 2021_

## Referee Comment (RC1)

**Review of de Vries et al ACP**

January 1, 2022

This article investigates the processes controlling the isotopic composition of upper-tropospheric water vapor and ice using an isotope-enabled simulation with a convection-permitting regional model. It identifies 5 main processes, as summarized in their last figure. Main strengths of this article are:

1. The convection-permitting simulation, combined with process-oriented diagnostics, allow for a very detailed process study.

2. This is the first study investigating the isotopic composition in the upper troposphere with such detail in a realistic setting.

This article is thus a significant contribution to the field. In addition, the article is well-written and well illustrated. For these reasons, I recommend publication of this study after some revisions.

**1 Major comments**

**1.1 Improve the evaluation section**

**1.1.1 Evaluation of the isotopic distribution**

The isotopic composition is evaluated using precipitation composition from the GNIP network. The GNIP stations are very sparse in this region. In addition, this allows to evaluate the composition only when it rains. The vapor composition would allow a more continuous evaluation is both space and time. Finally, the precipitation composition is strongly affected by post-condensation processes during rain fall. This does not tell anything about the realism of the simulation in the upper troposphere, which is the subject of this study. The vapor is actually what is most relevant to evaluate for this study, since this is what is ultimately transported to the upper-troposphere.

Therefore, I suggest to evaluate the isotopic simulation with respect to the water vapor composition. It is now available with good spatio-temporal coverage from satellite observations. For example, the IASI data is now available as handy 1x1° twice-daily maps, including for the period of interest ([Diekmann et al., 2021]).

This evaluation could come in addition or in replacement of the GNIP evaluation (which could go in appendix or SI).

In Fig 4a: plotting the water vapor $\delta^2 H$ would allow to directly compare with observations on the same plot. l 304: "not available at these time scales": the vapor composition is.

**1.1.2 Evaluation of the precipitation distribution**

High-resolution precipitation products are now available to rigorously evaluate precipitation simulation, for example IMERG ([Huffman et al., 2015]) or TRMM/GPM ([Huffman et al., 2007]).

The comparison of the simulated precipitation to the few GNIP stations in Fig 3j is very frustrating. Products are available that would give a much better view of the precipitation distribution and allow for a more rigorous evaluation.

Figs 4b and 4c: the simulation could also be directly compared to observations.

**1.1.3 Evaluation in the upper troposphere**

The study focuses on convective and microphysical processes in the upper troposphere. I'm aware that no isotopic product is available at this altitude with a good spatio-temporal coverage. But it would be useful for the reader to have at least some assessment of the realism of the simulation for convective and microphysical

processes. For example, what is the realism of the water budget in Fig 15? Satellite datasets are available to perform some basic evaluation of the distribution of the relative humidity with respect to ice and of ice water content in the upper troposphere, at least from a climatological point of view (e.g. . For example: MLS ([Livesey et al., 2006]), AIRS ([Fetzer et al., 2003, Read et al., 2007]), Cloudsat ([Austin et al., 2009]).

**1.2 Relevance of the water budget and isotopic processes for the transport of water vapor through the TTL**

Section 5.3: the added value of this section, and the connection with previous sections, are not clear to me.

What do we know about the realism of this simulated budget? How does it quantitatively compare with previous studies? To what extent is this budget subject to debate? Are there any direct observations for it?

What do we learn from Fig 15? If this Fig is enough to describe the water budget of the TTL, then why should we bother during all previous sections to investigate water isotopes? Wouldn't Fig 15 suffice by itself to solve all questions?

If observations for this budget are uncertain, then could water isotopic observations help to constrain it? If so, the connection with the previous sections would be clearer.

l 591-593: this is confusing. Only Fig 15a contributes to this debate. This article mainly contributes to a better understanding of the processes controlling the isotopic composition, but how is this understanding useful for quantifying the budget of the TTL?

Finally, my understanding is that the main debate is not about the ice water content in the TTL, but rather about water fluxes through the tropopause (e.g. [Bolot and Fueglistaler, 2021]).

Here are some suggestions:

- remove this sub-section

- use this water budget to support the interpretation of previous figures, especially Fig 13e. In this case, then Fig 15a could come just before or after Fig 13 to help interpret it.

- use this water budget to compare with observations, if any, to evaluate the realism of the simulation in the upper troposphere (see my comment 1.1.3). In this case, move it to the evaluation section.

- extend this sub-section to show the connection with previous sub-sections, e.g. could we reconstitute this budget just based on isotopic variables in the simulation?

**2 Minor comments**

- around l 12-15: it would be useful to tell the reader here what is the horizontal and vertical resolution of the simulation that is investigated in detail, and tell that it is convection-permitting. All the process analysis is done on this simulation, so describing the other simulations in the abstract is not so important.

- l 16-17: this sentence is confusing, especially "leading to". It looks like the injection of ice is what leads to the depletion of the water vapor. Rather, it is the condensation and deposition that leads to the depletion of water vapor.

- l 21: "statistical evaluation" is mysterious at this stage. Reword with something like "statistical analysis over a 1x1 spatial domain and over one month".

- l 26: "base of convective updrafts": this is confusing because we can imagine the base of convective updrafts at the lifting condensation level, well below the freezing level. Reword as "in the mid-troposphere", or give a more specific altitude range.

- l 54: "North Africa": this usually refers to North of the Sahara. Your domain rather corresponds to Western Africa, or to the Sahel.

- l 189: why switching off the shallow convective scheme as well?

- l 245-255: here the d-excess is evaluated. Yet, the d-excess is never discussed again in the process analysis. The d-excess is strongly influenced by post-condensation processes as rain falls. So the model-data agreement or disagreement for d-excess does not tell anything about the realism of the simulation in the upper-troposphere, which is the subject of this study. I suggest to remove these paragraphs, or move it to appendix or SI.

- l 247: how does condensation temperature evolves along air trajectories in Western Africa?

- l 260: "strongly changes": this looks tiny from Fig 3.

- l 291: these 4 phases are not obvious from Fig 4, especially from Fig 4a (black curve). Maybe the domain for the average is too large: it incorporates both the position of the ITCZ before and after the monsoon onset. So we cannot clearly see the monsoon onset when averaging over such a large domain. Maybe try 10N-20N?

- Fig 3: how are the GNIP stations ordered? By latitude? Longitude?

- l 307: why was this MCS chosen? At first, I thought that this MCS, which happens before the monsoon onset, would not be very representative of the full period. I would have expected that the drier air would lead to more ice sublimation, and thus higher $\Delta\delta^2 H_{ice}$, than for MCS during the monsoon period. But Fig 7 suggests that this is not the case. Why? Any comment on this?

- Fig 5c, and everywhere else: the color scale does not allow to distinguish the white regions due to low ice water content, and the white regions due to near zero disequilibrium. Could you for example put some yellow instead of white in the middle of the color bar?

- Fig 6: the numbers are quite small, I had difficulties to read them when printed.

- l 330, 352: I think it would be useful to refer to the appendix already here, or even before. As a reader, I found it very helpful to read the appendix before reading these sections. The appendix is advertised in l 364, but I think that if readers have been able to follow the text up to this line, they don't need the appendix any more. Many readers will need it before.

- l 322-364: All along the discussion of these processes, I was wondering to what extent they are consistent with previous studies. Which processes are new? Which ones are already well established? More references to previous studies would be useful here, to help the readers assess the contribution of this study compared to previous ones.

- l 365-374: This paragraph is crucial to show the representativeness of your case study for the whole period.

- l 376: why only July?

- section 4.3: do you still use the same spatio-temporal domain for this analysis?

- l 447-448: recall these process: condensation/deposition in updrafts and sublimation in downdrafts?

- Fig 8: Add some horizontal lines on fig a and b for 125, 200 and 450 hPa?

- Fig 9: why can't we see the positive vertical velocity anomaly for (1) on Fig 9b? Is it because $\delta^2 H_{ice}$ is also very high in convective downdrafts?

- Fig 10f: is there anything to tell about the super-saturation in strong updrafts?

- Fig 10f: strong updrafts above 200hPa are as dry as strong downdrafts. Why? And why isn't there any sublimation that would enrich the vapor, as in strong downdrafts?

- Fig 12: "black hatching": I cannot see it when printed. Try another color? Or wider rectangles?

- l 493: "mixed" is confusing, it recalls the mixed-phase clouds. Rather call it "outflow"?

- Fig 15: add horizontal lines to recall where the TTL is.

- l 585: add "idealized" in front of "large-eddy simulation". The limitation of previous modeling studies was their idealized configuration, not their resolution.

- l 696: why "and sublimation"? My understanding is that this section only discussed sedimentation, not sublimation?

**References**

[Austin et al., 2009] Austin, R. T., Heymsfield, A. J., and Stephens, G. L. (2009). Retrieval of ice cloud microphysical parameters using the cloudsat millimeter-wave radar and temperature. *Journal of Geophysical Research: Atmospheres*, 114(D8).

[Bolot and Fueglistaler, 2021] Bolot, M. and Fueglistaler, S. (2021). Tropical water fluxes dominated by deep convection up to near tropopause levels. *Geophysical Research Letters*, 48(4):e2020GL091471.

[Diekmann et al., 2021] Diekmann, C. J., Schneider, M., Ertl, B., Hase, F., García, O., Khosrawi, F., Sepúlveda, E., Knippertz, P., and Braesicke, P. (2021). The global and multi-annual musica iasi {H 2 O, $\delta$D} pair dataset. *Earth System Science Data*, 13(11):5273–5292.

[Fetzer et al., 2003] Fetzer, E., McMillin, L. M., Tobin, D., Aumann, H. H., Gunson, M. R., McMillan, W. W., Hagan, D. E., Hofstadter, M. D., Yoe, J., Whiteman, D. N., et al. (2003). Airs/amsu/hsb validation. *IEEE transactions on geoscience and remote sensing*, 41(2):418–431.

[Huffman et al., 2015] Huffman, G. J., Bolvin, D. T., Nelkin, E. J., and Tan, J. (2015). Integrated multi-satellite retrievals for gpm (imerg) technical documentation. *NASA/GSFC Code*, 612(47):2019.

[Huffman et al., 2007] Huffman, G. J., Bolvin, D. T., Nelkin, E. J., Wolff, D. B., Adler, R. F., Gu, G., Hong, Y., Bowman, K. P., and Stocker, E. F. (2007). The trmm multisatellite precipitation analysis (tmpa): Quasi-global, multiyear, combined-sensor precipitation estimates at fine scales. *Journal of Hydrometeorology*, 8(1):38–55.

[Livesey et al., 2006] Livesey, N. J., Van Snyder, W., Read, W. G., and Wagner, P. A. (2006). Retrieval algorithms for the eos microwave limb sounder (mls). *IEEE Transactions on Geoscience and Remote Sensing*, 44(5):1144–1155.

[Read et al., 2007] Read, W., Lambert, A., Bacmeister, J., Cofield, R., Christensen, L., Cuddy, D., Daffer, W., Drouin, B., Fetzer, E., Froidevaux, L., et al. (2007). Aura microwave limb sounder upper tropospheric and lower stratospheric h2o and relative humidity with respect to ice validation. *Journal of Geophysical Research: Atmospheres*, 112(D24).

---

## Referee Comment (RC2)

**Review of De Vries et al., submitted to ACP**

Title: Stable water isotope signals in tropical ice clouds in the West African monsoon simulated with a regional convection-permitting model

Manuscript no: acp-2021-902

**General comments :**

This manuscript investigates the processes setting the isotopic composition of vapor and ice in tropical ice clouds simulated by a regional convection-permitting model. This study is the first to use an isotope-embedded convection-permitting model for that purpose. The authors find five key processes in tropical ice clouds that can be distinguished based on their isotopic signature, and which are summarized in the last figure. I find that the manuscript is well written and constitutes an interesting contribution. I therefore recommend publication after minor revisions.

**Specific comments :**

The evaluation section is focused on the comparison of precipitation to GNIP observations. But, as the authors mention, the isotopic composition of precipitation is set by post-condensation processes, and therefore it doesn't bear a direct link to the isotopic composition of vapor and ice in the upper troposphere, which is the focus of the present paper. I think the authors should mention this more clearly.

In Figure 2(j), I suggest the authors use some satellite dataset of precipitation instead of GNIP to perform basic comparison with the model. One example would be TRMM/GPM. Alternatively, the authors could use the precipitation analysis from ERA5.

I have a comment concerning the budgets in the TTL at section 5.3 and the water budget displayed at Figure 15. While the partitioning of water discussed by the authors is interesting, it does not directly address the issue of the water budget in the TTL, since it does not address the underlying fluxes, or how the isotopic information gives specific insights on that issue. I suggest the authors merge this section with the previous one and use Figure 15 to help with the interpretation of Figure 13 instead.

**Technical corrections :**

Lines 168 – 175: "Fractionation is parameterized using equilibrium fractionation factors with respect to liquid water and ice, following Majoube (1971) and Merlivat and Nief (1967), respectively. Non-equilibrium fractionation effects occur, for instance, if the air is supersaturated with respect to ice, which is taken into account by a combined fractionation factor" I find the formulation a bit confusing as it seems to suggest that fractionation is parametrized overall as an equilibrium process, before mentioning the parametrization of non-equilibrium effects. I suggest reformulating: "Fractionation at thermodynamic equilibrium is parametrized using equilibrium fractionation factors […]. Non-equilibrium effects are taken into account by a combined fractionation factor […]. Such effects occur, for instance, if the air is supersaturated with respect to ice." I also suggest you mention that non-equilibrium effects arising from ice surface kinetics (Nelson, 2011) are neglected in the model.

Lines 280 – 282: "At this resolution, there is no clear difference between the parameterized and explicit convection setup as EXLP14 performs better than PAR14 in June, while no clear differences between both simulations emerge in July". Please repeat the resolution for clarity: "**At 14 km resolution**, there is no clear difference between the parameterized and explicit convection setup as EXLP14 performs better than PAR14 in June, while no clear differences between both simulations emerge in July"

Line 316: Typo. "Figure 4c shows the deviation of the isotopic composition of ice ($\delta$2Hice as in Fig. 4b) from the ice that would **form** from local vapour under equilibrium fractionation"

Around line 317: I suggest you mention that disequilibrium in ice can be produced both as a result of non-equilibrium conditions at fractionation and/or because of the lack of diffusive exchanges between vapor and ice, which allows enriched ice lofted from below to persist at higher levels in the atmosphere.

Lines 350 – 353: I realize that the model probably doesn't have a Wegener-Bergeron-Findeisen effect in mixed-phase cloud layers. Such an effect would also introduce disequilibrium in ice. This should be mentioned somewhere, probably in Section 2.1

Lines 463 – 465: "As already explained in Sect. 4.1, this signal stems from the lower equilibrium fractionation factor of condensation than that of vapour deposition." You could add "[…], **thus resulting in liquid water being isotopically lighter than ice**."

Lines 489 – 494 and thereafter: The formulation of ice cloud categories as "convective", "mixed" and "cirrus" is a bit misleading in my opinion. IWP discriminates between regimes over the entire depth of the troposphere, not individual clouds. For instance, in line 520, you mention "cirrus clouds" and "lower troposphere" in the same sentence, which contradicts the ISCCP classification of cirrus having cloud top pressure less than 440 hPa. I also think that the term "mixed" is misleading because it could be interpreted in the sense of "mixed-layer". I suggest you use the denominations "convective regime", "anvil regime" and "thin cirrus regime" when classifying the regimes by IWP, and consistently modify everywhere.

Lines 573 – 575: "In the upper troposphere and the lower TTL (125-200 hPa), deep convection contributes to more than 40 % to the total water budget" You should reformulate as "More than 40% of total water is within the deep convective regime." Your initial formulation could be interpreted as saying that total water even outside of deep convection bears a convective origin, which is not what you show here. Again, I think this section could be merged with the previous one to help explain the results of Figure 13.

Line 637: "Deep convection, although only occurring at about 3.8 % in time and space, contributes to about 40% of the total water budget" Same here, I would suggest: "Deep convection, although only occurring at about 3.8 % in time and space, **contains about 40% of total water**"

Lines 683 – 686: I suggest you write "alpha_eq(T) = **R_ice,eq** / R_vap > 1" to distinguish between the values of R_ice at a particular level and **R_ice,eq** entering the definition of alpha_eq. At line 686, you should write that R_{cloud ice} is **approximately equal** to alpha_eq R_vap, since kinetic effects can induce deviations, especially in strong updrafts where supersaturated conditions may prevail.

Line 697: Again, **R_ice is approximately equal to alpha_eq R_vap** since non-equilibrium conditions can occur, as you mention. Besides, this relationship applies here because you assume bulk equilibrium

between ice and vapor, as expected under in situ formation conditions. Otherwise, when ice crystals are grown from the nucleus, fractionation equilibrium only applies between the surface of ice crystals and ambient vapor, since diffusive exchanges cannot take place with the inner part of ice crystals. This condition of bulk equilibrium should be stated.

Figures 12, 13, 15: I would use the denominations "convective", "anvil" and "thin cirrus" for the type classification, and use "cloud regime regions" instead of "cloud type regions" in legend.

References:

Nelson, J. (2011). Theory of isotopic fractionation on facetted ice crystals. *Atmospheric Chemistry and Physics*, *11*(22), 11351–11360.

---

## Author Comment (AC1)

**Reply to reviewers' comments**

**Paper acp-2021-902**

**Stable water isotope signals in tropical ice clouds in the West African monsoon simulated with a regional convection-permitting model**

**by Andries Jan de Vries**

We thank all three reviewers for their thoughtful and constructive comments that help us to improve the manuscript. Based on the reviewers' suggestions, we implement several major and minor changes in the manuscript. The major changes are that we:

- Include a model evaluation against IASI satellite-based observations of $\delta^2H$ in tropospheric water vapour in the new section 3.1 following the suggestions of all three reviewers. In addition, we add GPM IMERG precipitation to the model evaluation based on precipitation;
- Remove the previous section 5.3 on the water budget of the TTL, and conclusions based on this section, from the manuscript, and use the information of previous Fig. 15 to support the analysis of section 5.2 following suggestions of rev. #1 and #3;
- Reduce the length of the manuscript by removing parts of the model evaluation, that is, d-excess in precipitation following the suggestion of rev. #1.

Below we provide a one-to-one response to all points raised by the reviewers. The reviewers' comments are in black and our replies in blue italics.

**Reply to comments from Reviewer 1 (RC1)**

*Before proceeding to the responses, we would like to note that the reviewer probably referred to the figure and line numbers as in the originally submitted manuscript. The editor asked for a first round of minor corrections before posting the preprint. Therefore, there may be an apparent discrepancy in our response when referring to figure and line numbers, for which we always used those as in the manuscript version that appeared in the discussion phase.*

**1 Major comments**

**1.1 Improve the evaluation section**

**1.1.1 Evaluation of the isotopic distribution**

The isotopic composition is evaluated using precipitation composition from the GNIP network. The GNIP stations are very sparse in this region. In addition, this allows to evaluate the composition only when it rains. The vapor composition would allow a more continuous evaluation is both space and time. Finally, the precipitation composition is strongly affected by post-condensation processes during rain fall. This does not tell anything about the realism of the simulation in the upper troposphere, which is the subject of this study. The vapor is actually what is most relevant to evaluate for this study, since this is what is ultimately transported to the upper troposphere. Therefore, I suggest to evaluate the isotopic simulation with respect to the water vapor composition. It is now available with good spatio-temporal coverage from satellite observations. For example, the IASI data is now available as handy 1x1° twice-daily maps, including for the period of interest ([Diekmann et al., 2021]). This evaluation could come in addition or in replacement of the GNIP evaluation (which could go in appendix or SI). In Fig 4a: plotting the water vapor $\delta^2H$ would allow to directly compare with observations on the same plot. l 304: "not available at these time scales": the vapor composition is.

*We agree with the reviewer that the current model evaluation using monthly GNIP data is limited as a result of the sparse station density and the coarse temporal (monthly) resolution of the observations. Following the reviewer's suggestion, we add a subsection (section 3.1 in the revised manuscript) with a comparison of the tropospheric water vapour isotopic composition from the three COSMO$_{iso}$ simulations against IASI data that is available twice per day at a 1x1 degree horizontal resolution at 4.2 km above sea level in cloud-free conditions*

*(Diekmann et al., 2021). When bearing in mind that we can only evaluate the model in cloud free conditions and when acknowledging that MCS systems cannot be expected to be simulated at the right time and place by any numerical model, we can only compare the model to the IASI data in a statistical sense. We do agree with the reviewer that cloud free water vapour isotope signals in the mid- and upper troposphere in the tropics most likely relate to the processes in cloudy updrafts, but we would like to point out that this relation is likely complex. This was nicely shown in a recent conference contribution using AIRS data (Risi et al. 2021) by relating the free tropospheric cloud free $d^2H$ to different characteristics of mesoscale convective organisation such as the number of storms and their duration. Therefore, a detailed evaluation of COSMO$_{iso}$ with IASI data allows to draw only limited conclusions on the model's ability to simulate delta values in tropospheric vapour and (convective) processes that influence the isotopic composition of water vapour. For this reason, we consider the GNIP observations as a useful and complementary component of our model evaluation. Even though GNIP observations are only available for a few stations and at monthly timescales, these observations contain integrated information of moist processes in the water cycle, including surface evaporation, condensation and deposition in convective updrafts, and reevaporation/sublimation of condensates below the clouds in precipitating regions. These processes are of direct interest to our study, see also the 5 processes related to ice cloud formation and decay, which form a central part of our study.*

*Preliminary results of our model comparison to IASI data shows that vapour in the model is more enriched as compared to the observation with about ~50 permil, whereby the vapour isotopes in the three model simulations are very similar to another. This comparison also shows that the model and observations follow a very similar tendency in time. This model evaluation concerns the water vapour in the middle troposphere (4220 m a.s.l.). Unfortunately, to the best of our knowledge, there are no vapour isotope observations available for the upper troposphere. Please, note that a more comprehensive comparison between the COSMO$_{iso}$ output, IASI observations and other modelling products is currently under way as a part of another study from Christopher Diekmann's PhD thesis.*

*To summarize, taking the above into consideration, we decided to include a combined model evaluation based on both the twice-daily IASI vapour isotopes and the monthly precipitation GNIP station data. We consider this combined model evaluation using IASI and GNIP observations as a useful and complementary approach, allowing us to cover different processes and aspects in the evaluation of our model simulations.*

*In addition, we will better emphasize in the revised manuscript that COSMO$_{iso}$ simulations have been evaluated against observations in several previous studies: Pfahl et al. (2012) showed a comparison against isotopes in six-hourly accumulated precipitation related to a North American cyclone; Aemisegger et al. (2015) against isotopes in hourly observations of vapour and precipitation during a cold frontal passage across Central Europe; Dütsch (2018) evaluated a 20-year simulation with GNIP data over Europe and, particularly relevant for this study, Dahinden et al. (2021) performed an extensive validation of simulated isotopes in the free troposphere above the eastern subtropical North Atlantic using observations from aircraft, surface and satellite remote sensing products. All these comparisons revealed in general a good agreement between observations and COSMO$_{iso}$ simulations.*

**1.1.2 Evaluation of the precipitation distribution**

High-resolution precipitation products are now available to rigorously evaluate precipitation simulation, for example IMERG ([Huffman et al., 2015]) or TRMM/GPM ([Huffman et al., 2007]). The comparison of the simulated precipitation to the few GNIP stations in Fig 3j is very frustrating. Products are available that would give a much better view of the precipitation distribution and allow for a more rigorous evaluation. Figs 4b and 4c: the simulation could also be directly compared to observations.

*Following the reviewer's suggestion, we add GPM IMERG satellite-based precipitation estimates to Fig. 3j (Fig. 3g in the revised manuscript). In addition, we consider adding GPM IMERG precipitation estimates to Fig. 4 to compare the monsoon evolution and motion of convective storms during the 2-month simulation period.*

**1.1.3 Evaluation in the upper troposphere**

The study focuses on convective and microphysical processes in the upper troposphere. I'm aware that no isotopic product is available at this altitude with a good spatio-temporal coverage. But it would be useful for the reader to have at least some assessment of the realism of the simulation for convective and microphysical processes. For example, what is the realism of the water budget in Fig 15? Satellite datasets are available to perform some basic evaluation of the distribution of the relative humidity with respect to ice and of ice water content in the upper

troposphere, at least from a climatological point of view (For example: MLS ([Livesey et al., 2006]), AIRS ([Fetzer et al., 2003, Read et al., 2007]), Cloudsat ([Austin et al., 2009]).

*Following the reviewer's comment 1.2 (below) and a similar comment from reviewer #3, we removed section 5.3 from the manuscript, and added Fig. 15 to the previous section 5.2 to clarify the vertical profiles of water species and their isotopic composition. Due to this change in the manuscript, and the already added model comparison to the IASI data, we consider it of less relevance to add a model evaluation that focuses on ice water content or relative humidity but misses isotope information.*

**1.2 Relevance of the water budget and isotopic processes for the transport of water vapor through the TTL**

Section 5.3: the added value of this section, and the connection with previous sections, are not clear to me. What do we know about the realism of this simulated budget? How does it quantitatively compare with previous studies? To what extent is this budget subject to debate? Are there any direct observations for it? What do we learn from Fig 15? If this Fig is enough to describe the water budget of the TTL, then why should we bother during all previous sections to investigate water isotopes? Wouldn't Fig 15 suffice by itself to solve all questions? If observations for this budget are uncertain, then could water isotopic observations help to constrain it? If so, the connection with the previous sections would be clearer. L 591-593: this is confusing. Only Fig 15a contributes to this debate. This article mainly contributes to a better understanding of the processes controlling the isotopic composition, but how is this understanding useful for quantifying the budget of the TTL? Finally, my understanding is that the main debate is not about the ice water content in the TTL, but rather about water fluxes through the tropopause (e.g. [Bolot and Fueglistaler, 2021]). Here are some suggestions:
• remove this sub-section
• use this water budget to support the interpretation of previous figures, especially Fig 13e. In this case, then Fig 15a could come just before or after Fig 13 to help interpret it.
• use this water budget to compare with observations, if any, to evaluate the realism of the simulation in the upper troposphere (see my comment 1.1.3). In this case, move it to the evaluation section.
• extend this sub-section to show the connection with previous sub-sections, e.g. could we reconstitute this budget just based on isotopic variables in the simulation?

*We agree with the reviewer that the previous section 5.3 and Fig. 15 failed to address the posed question on processes that regulate the TTL water budget. This point was also raised by reviewers #2 and #3. Following the reviewer's 2nd suggestion, which was also suggested by reviewer #3, we remove section 5.3 from the manuscript and use the information of Fig. 15 to support the analysis in section 5.2. In this context, Fig. 15 (also Fig. 15 in the revised manuscript) helps us to interpret and understand the vertical profiles of the water species and their isotopic composition in the different ice cloud regimes.*

**2 Minor comments**

• around l 12-15: it would be useful to tell the reader here what is the horizontal and vertical resolution of the simulation that is investigated in detail, and tell that it is convection-permitting. All the process analysis is done on this simulation, so describing the other simulations in the abstract is not so important.

*We rephrase the sentence as "We perform convection-permitting simulations using the regional isotope-enabled model COSMO$_{iso}$ at 7 km horizontal grid spacing and 60 model levels for the period of June-July 2016".*

• l 16-17: this sentence is confusing, especially "leading to". It looks like the injection of ice is what leads to the depletion of the water vapor. Rather, it is the condensation and deposition that leads to the depletion of water vapor.

*Thank you for this comment. We will rephrase the sentence along with other revisions in the abstract.*

• l 21: "statistical evaluation" is mysterious at this stage. Reword with something like "statistical analysis over a 1x1 spatial domain and over one month".

*We agree and correct this part of the sentence as "statistical analysis of isotope signals in the West African monsoon ice clouds for a one-month period".*

• l 26: "base of convective updrafts": this is confusing because we can imagine the base of convective updrafts at the lifting condensation level, well below the freezing level. Reword as "in the mid-troposphere", or give a more specific altitude range.

*We agree with the reviewer and change this part of the sentence to "the freezing of liquid water in the mixed-phase cloud layer above the freezing level in convective updrafts.". We also changed similar phrases at other places in the manuscript (sects. 4.1 and 6 and Appendix A).*

• l 54: "North Africa": this usually refers to North of the Sahara. Your domain rather corresponds to Western Africa, or to the Sahel.

*We agree and replace "North Africa" by "Sahel" at this and few other parts of the manuscript.*

• l 189: why switching off the shallow convective scheme as well?

*This has mainly a technical reason. Stable water isotope tracers are not implemented in the shallow convection scheme in COSMO$_{iso}$. We clarify this aspect in the manuscript by adding to the sentence "...as stable water isotope tracers are not implemented in the shallow convection scheme in COMSO$_{iso}$". In addition, previous analyses have shown that COSMO simulations at various resolutions do not necessarily provide more realistic precipitation patterns if shallow convection is turned on (Vergara-Temprado et al., 2020).*

• l 245-255: here the d-excess is evaluated. Yet, the d-excess is never discussed again in the process analysis. The d-excess is strongly influenced by post-condensation processes as rain falls. So the model data agreement or disagreement for d-excess does not tell anything about the realism of the simulation in the upper-troposphere, which is the subject of this study. I suggest to remove these paragraphs, or move it to appendix or SI.

*We agree with the reviewer's suggestion. To simplify the study, and to reduce the length of the manuscript and the model evaluation part in particular, we remove the discussion of d-excess in precipitation in section 3.1, including d-excess in Figure 2, and Tables 2 and 3, from the manuscript. Removal of d-excess from the analysis does not lead to substantial changes in the interpretation of the model evaluation.*

• l 247: how does condensation temperature evolves along air trajectories in Western Africa?

*This sentence is removed from the manuscript as a part of the above-mentioned revision that removed the d-excess part from the analysis.*

• l 260: "strongly changes": this looks tiny from Fig 3.

*In our opinion, the isotopic composition of precipitation changes much more strongly when switching off the convection scheme compared to changing the resolution in convection-permitting setup. Compare for example Fig. 2b to Fig. 2d (Fig. 2e in the originally submitted manuscript) versus Fig. 2d to Fig. 2f (previously Fig.2h in the originally submitted manuscript). Especially along the northern fringe of the tropical rain belt large changes are visible between the parameterized and convection-permitting model simulations. These "strong changes" also refer to the diurnal cycle of precipitation as shown in the cited studies in this sentence. We also looked at this aspect, but didn't include it in the analysis,. For these reasons, we consider it justified to speak about "strong changes".*

• l 291: these 4 phases are not obvious from Fig 4, especially from Fig 4a (black curve). Maybe the domain for the average is too large: it incorporates both the position of the ITCZ before and after the monsoon onset. So we cannot clearly see the monsoon onset when averaging over such a large domain. Maybe try 10N-20N?

*We agree that these four phases are not clearly visible in previous Fig. 3a. Therefore, we remove Fig. 3a, but keep Fig. 3b (latitude-time distribution) which in our opinion clearly shows these four different phases, including the monsoon onset. To strengthen the analysis, we also include GPM IMERG estimates to confirm the model simulated monsoon evolution and motion of the convective storm.*

• Fig 3: how are the GNIP stations ordered? By latitude? Longitude?

*As written in the caption of Fig. 2, the station locations are "ordered from west to east". For clarity we rephrase the caption as "ordered by longitude from west to east".*

• l 307: why was this MCS chosen? At first, I thought that this MCS, which happens before the monsoon onset, would not be very representative of the full period. I would have expected that the drier air would lead to more ice sublimation, and thus higher $\delta^2 H_{ice}$, than for MCS during the monsoon period. But Fig 7 suggests that this is not the case. Why? Any comment on this?

*This is indeed an interesting and well-made point by the reviewer. At an early stage of this study, we also considered this aspect, and to ensure that this MCS case study is also representative of storms that occur after the monsoon onset, we also looked into detail in an MCS case study in late July (24-25 July 2016). Qualitatively, we observed exactly the same isotope signatures in water vapour and ice which set the basis for the 5 presented processes related to the formation and decay of tropical ice clouds. In addition, it is indeed true that, as the reviewer suggests, MCSs in June have slightly more enriched ice ($\delta^2 H_{ice}$ >-64 ‰) and more positive disequilibrium in ice (> 400 ‰) than storms that occur in the 2^{nd} half of July (typically -192 ‰ < $\delta^2 H_{ice}$ < -128 ‰ and positive disequilibrium in ice > 300 ‰; see Fig. 6b,c). It is not clear if this results from the gradual depletion of water during the monsoon progress or reflects relatively weaker convective systems in moister air during a later stage of the monsoon. In any case, because the differences are relatively small, and qualitatively the same processes are observed, we chose the MCS on 15 June that was extraordinarily strong and therefore illustrates the identified processes best.*

• Fig 5c, and everywhere else: the color scale does not allow to distinguish the white regions due to low ice water content, and the white regions due to near zero disequilibrium. Could you for example put some yellow instead of white in the middle of the color bar?

*We thank the reviewer for this comment, and we improve the clarity of all figures that show disequilibrium in ice by changing the white colour for values from -10 to +10 ‰ to light yellow (Figs. 4c, 5d,h, 6c, 9e, 10c, and 16 of the initial manuscript).*

• Fig 6: the numbers are quite small, I had difficulties to read them when printed.

*We enlarge the numbers in Fig. 5h.*

• l 330, 352: I think it would be useful to refer to the appendix already here, or even before. As a reader, I found it very helpful to read the appendix before reading these sections. The appendix is advertised in l 364, but I think that if readers have been able to follow the text up to this line, they don't need the appendix anymore. Many readers will need it before.

*We thank the reviewer for this note and added the following sentence at the end of the first paragraph of section 4.1: "Appendix A presents detailed information on the definition and interpretation of the concept of disequilibrium in ice that is introduced in this study."*

• l 322-364: All along the discussion of these processes, I was wondering to what extent they are consistent with previous studies. Which processes are new? Which ones are already well established? More references to previous studies would be useful here, to help the readers assess the contribution of this study compared to previous ones.

*The five processes related to tropical ice clouds put forward in this study, have partially also been addressed by several foregoing stable water isotope-related studies. Our study aims to present an integral perspective on the formation and decay of these ice clouds from a water isotope perspective. To explicitly address this aspect (i.e. what previous studies have already shown and what our work adds to that) we provide the following text in the "summary and conclusions" section, which we consider as the most relevant and appropriate place to discuss this aspect:*

*"Previous studies have addressed several of these processes from a stable water isotope perspective. Risi et al. (2008a) and Bony et al. (2008) pointed to the relevance of convective processes that affect the isotopic composition of water vapour, including condensate lofting, rain evaporation, unsaturated downdrafts, and the isotopic exchange between droplets and the tropospheric environment. Bolot et al. (2013) used a conceptual model to address the cloud microphysical processes that affect the water isotopes such as condensation, freezing and deposition in convective updrafts. Blossey et al. (2010) showed that sublimation of convectively lofted ice can enrich TTL vapour, while the fractionation of in situ ice formation can lead to the isotopic depletion of TTL vapour. In this study, we present an integral perspective on these ice cloud processes for which the concept of disequilibrium in ice (see Appendix A for more information) appears particularly useful."*

*Please, note that the introduction more extensively discusses previous studies that addressed ice clouds and convective processes from a stable water isotope perspective.*

• l 365-374: This paragraph is crucial to show the representativeness of your case study for the whole period.

*To emphasise this aspect, we replace the phrase* "are not exceptional, but typical" *by* "typical and representative".

• l 376: why only July?

*There are two specific reasons to focus on only July. We provide a clarification in the text and add the sentence in section 4.2 "The choice to use data for this one-month period only instead of the full two-month simulation period aims to circumvent the potential influence of systematic differences in stable water isotope signals before and after the monsoon onset (Risi et al., 2008) and the gradual depletion of stable water isotopes during the progress of the monsoon (Risi et al., 2010), see also Fig. 2 of the revised manuscript".*

• section 4.3: do you still use the same spatio-temporal domain for this analysis?

*Yes, we do. To clarify this, we add the sentence "The remainder of this study uses the model output for this region of interest and specific time period." in section 4.2 since this domain and period is used throughout the remainder of the study (also section 5). To avoid repetition, we remove a similar sentence later on in the manuscript, in the first paragraph of section 5.1.*

• l 447-448: recall these processes: condensation/deposition in updrafts and sublimation in downdrafts?

*We rephrase the ending of the sentence as "… and the moist processes that take place in deep convection, that is, condensation and deposition in updrafts and sublimation in downdrafts.".*

• Fig 8: Add some horizontal lines on fig a and b for 125, 200 and 450 hPa?

*Thank you for these suggestions. We add horizontal lines to demarcate the mixed-phase cloud layer (at T=0°C and T=-38°C) and tropopause cold point (~ 100 hPa), and grey shading for the TTL with the lower and upper bounds at 150 and 70 hPa in Fig. 7.*

• Fig 9: why can't we see the positive vertical velocity anomaly for (1) on Fig 9b? Is it because δ²H$_{ice}$ is also very high in convective downdrafts?

*Yes, we presume this is the case. Strong downdrafts also can have very enriched ice as also shown in Fig. 9c. For this reason, we explicitly depict the water species and their isotopic composition as a function of vertical motion in Fig. 9.*

• Fig 10f: is there anything to tell about the super-saturation in strong updrafts?

*We also thought about this aspect, in particular, why the relative humidity over ice falls below 100% in the upper troposphere (p < 225 hPa) for the strongest updrafts (w > 5 m s⁻¹). We speculate this may be due to overshooting convection that may become unsaturated due to entrainment of dry surrounding air. Since we prefer to avoid speculations, and since this aspect is not of central importance for the main results in this study, we prefer not to elaborate on this aspect in this manuscript.*

• Fig 10f: strong updrafts above 200hPa are as dry as strong downdrafts. Why? And why isn't there any sublimation that would enrich the vapor, as in strong downdrafts?

*For the first question, please, see the response to the minor comment above. For the second question, we have at present no explanation.*

• Fig 12: "black hatching": I cannot see it when printed. Try another color? Or wider rectangles?

*We plot higher bars that indicate the ice cloud regimes in which the black hatching appears now clearer.*

• l 493: "mixed" is confusing, it recalls the mixed-phase clouds. Rather call it "outflow"?

*We thank the reviewer for this comment and agree that the usage of this term is confusing. This aspect is also raised by rev #3. We rename the "mixed" clouds as "thick cirrus" throughout the manuscript.*

• Fig 15: add horizontal lines to recall where the TTL is.

*We added horizontal lines to indicate the position of the TTL as well as horizontal lines to indicate the cold point tropopause, and the liquid, mixed-phase, and ice cloud layers.*

• l 585: add "idealized" in front of "large-eddy simulation". The limitation of previous modeling studies was their idealized configuration, not their resolution.

*We added "idealized" in front of "large-eddy simulation".*

• l 696: why "and sublimation"? My understanding is that this section only discussed sedimentation, not sublimation?

*Sedimentation and sublimation processes are hard to disentangle. In fact, when ice is sedimenting, it will sublimate if $RH_{ice}$ is below 100%, which is almost always the case away from strong convective updrafts (see e.g. Fig. 9f). The combined process of sedimentation and sublimation is reflected by slightly negative disequilibrium in ice in regions with only weak vertical motion, see e.g. Fig.5h, 6c, and 9c. Process #3 differs substantially from process #4 as process #4 occurs in strong downdrafts where the sublimation of ice strongly enriches the environmental vapour. See for example labels "3" and "4" in Fig. 9d,e. For these reasons, we consider it most accurate to refer to "sedimentation and sublimation" for process #3 and to "sublimation in convective downdrafts" for process #4.*

**Reply to comments from Reviewer 2 (RC3)**

**Main point:**

Clearly, an upper troposphere (lower stratosphere) evaluation is lacking, given that the study is focused on these altitudes and that data is available there. Satellite data products, IAGOS-CARIBIC data and/or other in situ measurements (I give some examples below) could be used for that. In consequence of this, the paper will become even longer than it already is and my advice to act contrary to that would be to cut the paper after section 4. In my opinion, section 5 is a little half-baked and has much more potential than what is shown here, i.e. potentially in a separate paper. For example, for stratospheric water vapour budgeting, it is so important how much of the ice is sedimenting out of the UTLS again after the strong convective event and I guess this could be analysed well with this tool. Moreover, some more advanced estimation could be done on how strongly this process could affect the UTLS globally and annually (I write some more about this in the minor points). Section 4 can be shortened or partly moved to a supplement. The appendix can then also be moved to the supplement.

*Following the suggestion of the reviewer, we added a model evaluation of the isotopic composition of water vapour against the IASI satellite data (section 3.1 in the revised manuscript). This observation-based product contains water vapour and $\delta^2H$ measurements at a 1-degree regular grid at 4.2 km above sea level (Diekmann et al., 2021). In this evaluation we look at the spatial distribution and temporal evolution of vapour isotopes in the three model simulations and in IASI. We also would like to mention here that a more thorough comparison of the IASI observations to our and other model simulations is currently underway in a separate study resulting from Christopher Diekmann's PhD thesis.*

*This comparison focuses on the mid troposphere (4220 m a.s.l.) since the IASI estimates are best for this part of the atmosphere. To the best of our knowledge, also other satellite-based measurements provide the most reliable estimates for the mid-troposphere and not for the upper troposphere. The IAGOS-CARIBIC data has unfortunately no flight data available for our simulation period June-July 2016, and moreover, has only a hand full of flights in other years over our region of interest (flights between Germany and South Africa) in the winter season. For this reason, we cannot include IAGOS-CARIBIC data in our study.*

*We agree with the reviewer that section 5.3 of the previously submitted manuscript failed to address the posed question on the TTL water budget. This point was also raised by reviewers #1 and #3. Therefore, we removed section 5.3 from the manuscript, and used the previous Fig. 15 in section 5.2 to clarify the vertical profiles of water species and their isotopic composition (previous Figs. 12 and 13), which also follows the suggestions of reviewers #1 and #3.*

*Furthermore, we shortened our manuscript, and section 3 in particular, by removing the analysis of d-excess in precipitation from the manuscript.*

*Finally, we appreciate the reviewer's suggestion to use our model simulations to provide estimations of how tropical convection can affect the UTLS water vapour. We keep this suggestion in mind as it could be part of a follow-up study in which we could use fluxes of water vapour and ice to estimate how tropical convection affects the water vapour in this region of the atmosphere. As the reviewer also mentioned, including such an analysis in the present paper would make the manuscript too long.*

**Minor points:**
- The manuscript is full of long halting sentences that makes reading sometimes a little cumbersome. I suggest to go through the entire paper again in order to split these long sentences.

*We thank the reviewer for pointing this out. To improve the readability, we shortened several sentences in the manuscript, split up sentences, or removed long sentences, including lines 37-40, 298-300, 302-305, and 585 of the initial manuscript.*

- L9: Change 'predictions' to 'projections

*Corrected.*

- L16-20: For the abstract it would be enough to summarise this to one more general sentence.

*We will take this comment in to consideration along with several other revisions of the abstract.*

- Remove the last sentence of the abstract.

*We assume that the concluding sentence of the abstract was too general with having a global implication based on a regional modelling study. Therefore, we rephrase "the Earth's water cycle" as "the West African monsoon water cycle". We consider it important to conclude the abstract with the most important implication of the study.*

- L40: Solomon et al. 2010 Science (10.1126/science.1182488) should be mentioned here too.

*Corrected.*

- L175-180: What do you assume/use as delta values for the ocean surface?

*For the ocean we use a slightly enriched value of $\delta^{18}O = 1$ ‰ and $\delta^2H = 1$ ‰, following Pfahl et al. (2012).*

- L209: What variables do you nudge? And you do that within your domain, right? Not only at the boundaries?

*Within the domain we applied a spectral nudging of the horizontal wind field (u and v) above 850 hPa, as written in this sentence. For clarity we add the phrase "Within the model domain …" to the sentence.*

- L254 and everywhere throughout the paper: Remove "Interestingly"! Everything you write should be interesting of some sort, hence there is no reason whatsoever to emphasise some bits to be interesting.

*We removed the word "interesting" here and elsewhere.*

- L276-284: But according to the errors in the tables, precipitation seems to be best represented in the PAR14 simulation. Can you elaborate on that?

*For June the precipitation amounts are indeed better represented in the PAR14 simulation compared to the EXPL14 and EXPL7 simulations according to the root-mean-square-error (RMSE) and mean error indices. This is not surprising considering the fact that simulations with parameterized convection produce a more "smeared-out" precipitation pattern in contrast to convection-permitting simulations that simulate explicitly local convective precipitation imprints from strong local convective storms that are more realistic but typically not exactly at the right time and right place. As a consequence, when using grid-point based metrices like the RMSE, precipitation in the parameterized convection simulations often obtain better scores than convection-permitting simulations even though the nature of precipitation is worse represented (e.g. the diurnal cycle of precipitation). This is well-known in the verification literature as the double-penalty problem and object-based verification techniques have been developed to alleviate this issue (Wernli et al. 2008). Considering these facts, it is very interesting to see that the isotopic composition of precipitation actually is equally good (EXPL14) or improved (EXPL7) when switching of the convection scheme, further supporting our choice to use the convection-permitting simulation for the analysis in sections 4 and 5.*

- L303-306: Can you provide the correlation coefficient between precip and $\delta^2$H for the time series and the different phases. I assume the correlation will be much greater in phase 2 than in phase 1, that would strengthen your point and make this part more quantitative.

*The time series of precipitation and its isotopic composition have been removed from the paper (previously Fig. 3a).*

- L384: Steinwagner et al. (2010) could be cited here again too.

*Corrected.*

- L385-395: In this context, consider also the studies by Notholt et al. (2010), Coffey et al. (2006) JGR, 111:D14313, Sayres et al. (2010) JGR, 115:D00J20 and IAGOS-CARIBIC data, but that should anyway be extended by a proper UTLS evaluation of your simulations (see main point).

*We added the observed range of dD values as presented by Sayres et al. (2010): "... , values between –650 and –400 ‰ from a flight campaign out of Costa Rica in August 2007 (Sayres et al., 2010), ...".*

*The measurements of Notholt et al. (2010), shown in their Fig. 1, are well outside the tropics (~33N), while the results in Coffey et al. (2006) are slightly confusing as the text (paragraph 26 of their page 6) describes d$^2$H values of –460 +/-21 permil in the tropics, while the corresponding Fig. 7 shows values in the range of –500 to –600 ‰.*

- L582: Change title to "Summary and conclusions", because that is what it is.

*We agree and corrected this.*

- L619: Please state here that disequilibrium in water was already used (by Aemisegger et al. (2015) and Graf et al. (2019), as you state in L320) and now the idea was transferred to ice. As it is formulated now it seems like the entire idea is new.

*We correct the sentence as follows: "In this study, we present an integral perspective on these ice cloud processes for which the concept of disequilibrium in ice (see Appendix A for more information) appears particularly useful."*

- L636-637: It would be great to have a some more quantitative idea of what this means globally (or for the tropics) and annually, but as I state in the main point, this could be removed completely and taken up properly in a separate paper.

*We agree that this is a very interesting aspect suitable for a follow-up study. However, we do not see a clear and straightforward way how to extrapolate these findings based on our 2-month regional model simulations to global and annual scales.*

- L640: As I state in the main point, this is a very important point, in particular for budgeting of stratospheric water vapour. It is a little disappointing that this is not addressed more clearly here. But the paper as it stands

is already fine, and this point goes too far, hence the suggestion for a separate paper, taking up section 5 of that one.

*We thank the reviewer for this note and will keep the suggestion in mind to address this aspect of TTL water budgeting for a follow-up paper, please, see the comment above.*

- L642-643: That is one good way, but I think also with this model here you could already start to analyse this process to some degree. It could be done in various regions (including Asian and American monsoon), and compared to regional models that do not resolve convection and to global models (like by Eichinger et al. 2015) to evaluate the differences in various regions in ice lofting/overshooting, and the effect on stratospheric water vapour and delta values.

*Please, see the response to the previous comment.*

- L645-654: This paragraph is misplaced here. It should appear in a discussion section or along with the results.

*We think that it is a matter of taste where to mention these caveats and we prefer to not move this paragraph to another section.*

- Fig. 2: I have not seen where you clarify these station names. If you haven't, please do so. Moreover, enlarge the axis descriptions in panels m-o.

*We add the station names of these acronyms in the caption of the figure.*

- Fig. 5c and g: Use a colour blind-friendly colour bar!

*We are aware of the guidelines that recommend using colour blind-friendly colour bars (i.e. avoiding the use of red and green colours). Accordingly, we have tested several colour bars, please, see the examples below (using Figure 5 of the initial manuscript). However, with these colour bars another problematic issue arises. Due to the rather similar colours in the bar, it becomes very difficult to retrieve specific values from the figures to which we refer throughout the manuscript. Furthermore, we would like to point out that there are many different types of colour-blindness and that it is not trivial to find one that suits all of these. For these reasons, we prefer to use the originally used colour bar. Also, in terms of aesthetics we clearly prefer the originally used colour bar.*

[Figure]

*Figure 5 with the originally used colour bar, with green and red colours, but allowing to retrieve all values from the figure.*

[Figure]

*Figure 5 with a colour blind friendly colour bar from Scientific colour maps 7.0 (https://zenodo.org/record/5501399#.YhIzVS8w2qk) with purple to yellow sequential colours, but lacking the possibility to retrieve values from the figure.*

[Figure]

*Figure 5 with another colour blind friendly colour bar from Scientific colour maps 7.0 (https://zenodo.org/record/5501399#.YhIzVS8w2qk) with sequential colours, but lacking the possibility to retrieve values from the figure.*

- Fig. 9d, f, Fig. 10 b, Fig. 15: Use a colour blind-friendly colour bar. No green and red!

*Please, see the response to the previous comment. Also, we tested the green and red colours used in Fig. 15 of the previous manuscript which showed that the colours are different also for colour-blindness.*

- Fig. 12 and 13 could be combined.

*We thank the reviewer for pointing to this possibility. Actually, we had Figs. 12 and 13 combined in the originally submitted manuscript that was subject of early corrections. For the final version that appeared in preprint, we decided to split this figure in two separate figures to avoid a very large figure with 6 panels that are based on different data. Please, note also that the vertical range over pressure levels are different in these two figures and that Fig. 12 only includes grid points where ice ≥ 0.0001 mg kg⁻¹, while Fig. 13 includes all grid points in the region of interest.*

**Technical issues**
- L15: explores -> helps exploring

*We will take this suggestion into consideration for the revised abstract.*

- L17: ... leading to isotopic depletion of water vapour within....

*We rephrased the sentence, and also included the word "isotopic" in this rephrasing.*

- L38: ...stratosphere. This is a topic of ...

*Corrected.*

- L39: ...water vapour ...

*Corrected.*

- L41: regions

*Corrected.*

- L182: western

*Corrected.*

- L186: vertical model levels -> model levels in the vertical (This is wrong in so many papers, but it doesn't become correct that way)

*Corrected.*

- L222: ... output to these observations...

*Corrected.*

- L223: ... simulations. The intention ...

*Corrected.*

- L224: validation -> evaluation

*Corrected.*

- L257: how -> the way

*Corrected.*

- L259: 'Relevant here is that...' please rephrase

*We rephrased the sentence as "The relevant point here is that …".*

- L261: 'Consistent with previous studies, switching off...' please rephrase

*We rephrased the sentence as "Switching off the parameterization scheme strongly changes the hydrological cycle, consistent with previous studies that focused on the WAM region …".*

- L265: over -> at, over -> in

*We corrected the first over as "along" and the second as "in".*

- L302 and in many other occasions: Remove "very".

*At this and several other occasions we removed "very".*

- L295: its -> their

*Corrected.*

- L397: single -> specific

*Corrected.*

- L437-439 Remove ", which is ....Not surprisingly,"

*Corrected.*

- L445: corresponds

*Corrected.*

- L551: Remove: "Importantly"

*Done.*

- L584 and L606: It may help many readers if the meaning of the abbreviations WAM and MCS are repeated again in the conclusions section.

*Corrected.*

- L638: Remove 'By large'

*Done.*

- Fig. 5: Enlarge axis descriptions, panel titles and colour palettes including numbers there.

*Done.*

- Fig. 7c-e: Add another legend (in particular due to red dashed line).

*Done.*

- Fig. 11: Enlarge axis descriptions, panel titles and colour palettes including numbers
There.

*Done.*

**Response to comments of reviewer #3 (RC2)**

Specific comments:

The evaluation section is focused on the comparison of precipitation to GNIP observations. But, as the authors mention, the isotopic composition of precipitation is set by post-condensation processes, and therefore it doesn't

bear a direct link to the isotopic composition of vapor and ice in the upper troposphere, which is the focus of the present paper. I think the authors should mention this more clearly.

*In the revised manuscript we include a comparison of the isotopic composition of vapour from the different COSMO$_{iso}$ simulations to IASI satellite data (see section 3.1 and Fig. 2 of the revised manuscript and our responses to the major comments of reviewers 1 and 2).*

In Figure 2(j), I suggest the authors use some satellite dataset of precipitation instead of GNIP to perform basic comparison with the model. One example would be TRMM/GPM. Alternatively, the authors could use the precipitation analysis from ERA5.

*We included GPM IMERG monthly precipitation amounts in Fig. 2j and Fig. S1 of the initial manuscript (Fig. 3g and Fig. S1 in the revised manuscript). This shows that there is a good qualitative agreement between precipitation from the model simulations and the satellite-based observations.*

I have a comment concerning the budgets in the TTL at section 5.3 and the water budget displayed at Figure 15. While the partitioning of water discussed by the authors is interesting, it does not directly address the issue of the water budget in the TTL, since it does not address the underlying fluxes, or how the isotopic information gives specific insights on that issue. I suggest the authors merge this section with the previous one and use Figure 15 to help with the interpretation of Figure 13 instead.

*We thank the reviewer for pointing out this aspect that was also identified by the other reviewers. Following the reviewer's suggestion, we remove the previous section 5.3 from the manuscript and include the information from Figure 15 in section 5.2 to clarify the content of this section as well as the information shown in Figs. 12 and 13. At all places in the manuscript we removed all inferences suggesting we studied the TTL water budget based on this analysis.*

Technical corrections:
Lines 168 – 175: "Fractionation is parameterized using equilibrium fractionation factors with respect to liquid water and ice, following Majoube (1971) and Merlivat and Nief (1967), respectively. Non-equilibrium fractionation effects occur, for instance, if the air is supersaturated with respect to ice, which is taken into account by a combined fractionation factor" I find the formulation a bit confusing as it seems to suggest that fractionation is parametrized overall as an equilibrium process, before mentioning the parametrization of non-equilibrium effects. I suggest reformulating: "Fractionation at thermodynamic equilibrium is parametrized using equilibrium fractionation factors […]. Non-equilibrium effects are taken into account by a combined fractionation factor […]. Such effects occur, for instance, if the air is supersaturated with respect to ice." I also suggest you mention that non-equilibrium effects arising from ice surface kinetics (Nelson, 2011) are neglected in the model.

*We corrected the text as suggested, including the notion that non-equilibrium effects arising from ice surface kinetics are neglected in the model.*

Lines 280-282: "At this resolution, there is no clear difference between the parameterized and explicit convection setup as EXLP14 performs better than PAR14 in June, while no clear differences between both simulations emerge in July". Please repeat the resolution for clarity: "**At 14 km resolution**, there is no clear difference between the parameterized and explicit convection setup as EXLP14 performs better than PAR14 in June, while no clear differences between both simulations emerge in July"

*Corrected.*

Line 316: Typo. "Figure 4c shows the deviation of the isotopic composition of ice ($\delta$2Hice as in Fig. 4b) from the ice that would **form** from local vapour under equilibrium fractionation".

*Corrected.*

Around line 317: I suggest you mention that disequilibrium in ice can be produced both as a result of non-equilibrium conditions at fractionation and/or because of the lack of diffusive exchanges between vapor and ice, which allows enriched ice lofted from below to persist at higher levels in the atmosphere.

*We added the following sentence to the first paragraph of section 4.1 "Disequilibrium in ice can result from non-equilibrium conditions during ice formation, and, of primary interest in this study, the absence of equilibrium*

*exchanges between vapour and ice allows for a preservation of the isotope signal of convectively lofted ice at higher altitudes in the atmosphere".*

Lines 350 – 353: I realize that the model probably doesn't have a Wegener-Bergeron-Findeisen effect in mixed-phase cloud layers. Such an effect would also introduce disequilibrium in ice. This should be mentioned somewhere, probably in Section 2.1.

*The model does represent the Wegener-Bergeron-Findeisen effect by formulating the depositional growth of cloud ice as non-equilibrium process. We include the following sentences in section 2.1:*

*"Non-equilibrium effects arising from ice surface kinetics (Nelson, 2011) are neglected in the model. The Wegener-Bergeron-Findeisen effect is represented in the model and the non-equilibrium effects during depositional growth of ice are accounted for.".*

Lines 463 – 465: "As already explained in Sect. 4.1, this signal stems from the lower equilibrium fractionation factor of condensation than that of vapour deposition." You could add "[…], **thus resulting in liquid water being isotopically lighter than ice**."

*We added the following phrase to the sentence "deposition thus resulting in liquid water being isotopically lighter than ice that would form directly from the vapour phase".*

Lines 489 – 494 and thereafter: The formulation of ice cloud categories as "convective", "mixed" and "cirrus" is a bit misleading in my opinion. IWP discriminates between regimes over the entire depth of the troposphere, not individual clouds. For instance, in line 520, you mention "cirrus clouds" and "lower troposphere" in the same sentence, which contradicts the ISCCP classification of cirrus having cloud top pressure less than 440 hPa. I also think that the term "mixed" is misleading because it could be interpreted in the sense of "mixed-layer". I suggest you use the denominations "convective regime", "anvil regime" and "thin cirrus regime" when classifying the regimes by IWP, and consistently modify everywhere.

*We agree entirely with the reviewer and implement the suggested changes: (1) we revise the text by writing "ice cloud regimes" rather than "ice cloud categories", (2) we rename the three regimes as "convective", "thick cirrus", and "thin cirrus", which is also following the exact same naming as used by Turbeville et al. (2021), and (3) we correct the use of "lower troposphere" by "middle troposphere" (lines 519 and 523) when discussing ice clouds in pressure layers of approximately 400 to 600 hPa.*

Lines 573 – 575: "In the upper troposphere and the lower TTL (125-200 hPa), deep convection contributes to more than 40 % to the total water budget". You should reformulate as "More than 40% of total water is within the deep convective regime." Your initial formulation could be interpreted as saying that total water even outside of deep convection bears a convective origin, which is not what you show here. Again, I think this section could be merged with the previous one to help explain the results of Figure 13.

*We thank the reviewer for this constructive feedback. We remove section 5.3 from the manuscript and move the previous Fig. 15 to section 5.2 to help clarifying the information in Figures 12 and 13 of the initial manuscript (Fig. 13 and 14 of the revised manuscript). Also, we add a paragraph at the end of section 5.2 that discusses the figure in a manner to place the previous discussion in context of the water partitioning across the four cloud regimes. This underlines the importance of deep convection for TTL water vapour through convective ice lofting. In our revised text we carefully avoid any inferences to the water budget as we indeed have not computed fluxes that would be needed to address the TTL water budget.*

Line 637: "Deep convection, although only occurring at about 3.8 % in time and space, contributes to about 40% of the total water budget" Same here, I would suggest: "Deep convection, although only occurring at about 3.8 % in time and space, **contains about 40% of total water**"

*Corrected in the text of section 5.2, please, see the response to the previous comment.*

Lines 683 – 686: I suggest you write "alpha_eq(T) = **R_ice,eq** / R_vap > 1" to distinguish between the values of R_ice at a particular level and **R_ice,eq** entering the definition of alpha_eq. At line 686, you should write that R_{cloud ice} is **approximately equal** to alpha_eq R_vap, since kinetic effects can induce deviations, especially in strong updrafts where supersaturated conditions may prevail.

*We thank the reviewer for these thoughtful comments and implemented both changes as suggested.*

Line 697: Again, **R_ice is approximately equal to alpha_eq R_vap** since non-equilibrium conditions can occur, as you mention. Besides, this relationship applies here because you assume bulk equilibrium between ice and vapor, as expected under in situ formation conditions. Otherwise, when ice crystals are grown from the nucleus, fractionation equilibrium only applies between the surface of ice crystals and ambient vapor, since diffusive exchanges cannot take place with the inner part of ice crystals. This condition of bulk equilibrium should be stated.

*We corrected the text as suggested and replaced "=" by "≈" in both equations of isotope ratios and disequilibrium in ice, and further added the phrase "assuming bulk equilibrium between vapour and ice" to this sentence.*

Figures 12, 13, 15: I would use the denominations "convective", "anvil" and "thin cirrus" for the type classification, and use "cloud regime regions" instead of "cloud type regions" in legend.

*Consistent with the changes in the text of section 5, we adopted the denominations "convective", "thick cirrus", and "thin cirrus". (We prefer here the more general term "thick cirrus" over "anvil" as this regime includes a multitude of different cirrus, including remnants of convective storms, juvenile convective systems and thicker cirrus shields & convective outflow). Also, following the suggestion of the reviewer, we replace "cloud type regions" by "cloud regimes" in the figure captions.*

**References**

Aemisegger, F., Spiegel, J. K., Pfahl, S., Sodemann, H., Eugster, W., and Wernli, H.: Isotope meteorology of cold front passages: A case study combining observations and modeling, Geophys. Res. Lett., 42, 5652–5660, https://doi.org/10.1002/2015GL063988, 2015.

Dahinden, F., Aemisegger, F., Wernli, H., Schneider, M., Diekmann, C. J., Ertl, B., Knippertz, P., Werner, M., and Pfahl, S.: Disentangling different moisture transport pathways over the eastern subtropical North Atlantic using multi-platform isotope observations and high-resolution numerical modelling, Atmos. Chem. Phys., 21, 16319–16347, https://doi.org/10.5194/acp-21-16319-2021, 2021.

Diekmann, C. J., Schneider, M., Ertl, B., Hase, F., Garcia, O., F., Khosrawi, F., Sepulveda, E., Knippertz, P., and Braesicke, P.: The global and multi-annual MUSICA IASI {$H_2O$, $\delta D$} pair dataset, Earth Syst. Sci. Data, 13, 5273-5292, 2021.

Dütsch, M., Pfahl, S., Meyer, M., and Wernli, H.: Lagrangian process attribution of isotopic variations in near-surface water vapour in a 30-year regional climate simulation over Europe, Atmos. Chem. Phys., 18, 1653–1669, https://doi.org/10.5194/acp-18-1653-2018, 2018.

Pfahl, S., Wernli, H., and Yoshimura, K.: The isotopic composition of precipitation from a winter storm – a case study with the limited-area model COSMOiso, Atmos. Chem. Phys., 12, 1629–1648, https://doi.org/10.5194/acp-12-1629-2012, 2012.

Risi, C., Bony, S., Vimeux, F., Descroix, L., Ibrahim, B., Lebreton, E., Mamadou, I., and Sultan, B.: What controls the isotopic composition of the African monsoon precipitation? Insights from event-based precipitation collected during the 2006 AMMA field campaign, Geophys. Res. Lett., 35, L24808, doi:10.1029/2008GL035920, 2008.

Risi, C., Bony, S., Vimeux, F., Frankenberg, C., Noone, D., and Worden, J.: Understanding the Sahelian water budget through the isotopic composition of water vapor and precipitation, J. Geophys. Res.-Atmos., 115, D24110, https://doi.org/10.1029/2010JD014690, 2010.

Risi, C., Roca, R. Fiolleau, T., Vimeux, F., Etienne, J., and Worden, J., Impact of convective organization on tropospheric humidity and isotopic composition, Workshop on "Water isotopes from weather to climate", 15-17 Nov., online, 2021.

Sayres, D. S., Pfister, L., Hanisco, T. F., Moyer, E. J., Smith, J. B., St. Clair, J. M., O'Brien, A. S., Witinski, M. F., Legg, M., and Anderson, J. G.: Influence of convection on the water isotopic composition of the tropical

tropopause layer and tropical stratosphere, J. Geophys. Res.-Atmos., 115, D00J20, doi:10.1029/2009JD013100, 2010.

Turbeville, Nugent, J. M., Ackerman, T. P., Bretherton, C. S., and Blossey, P. N.: Tropical cirrus in global storm-resolving models. Part II: Cirrus life cycle and top-of-atmosphere radiative fluxes, submitted to Earth and Space Science, https://doi.org/10.1002/essoar.10507887.2, 2021.

Vergara-Temprado, J., Ban, N., Panosetti, D., Schlemmer, L., and Schär, C.: Climate models permit convection at much coarser resolutions than previously considered, J. Climate, 33, 1915–1933, https://doi.org/10.1175/JCLI-D-19-0286.1, 2019.

Wernli, H., Paulat, M., Hagen, M., and Frei C.: SAL-A novel quality measure for the verification of quantitative precipitation forecasts, Mon. Wea. Rev., 136, 4470-4487, 2008.

---

## Referee Report (RR1)

**Review of De Vries et al., submitted to ACP**

Title: Stable water isotope signals in tropical ice clouds in the West African monsoon simulated with a regional convection-permitting model

Manuscript no: acp-2021-902

Iteration no: 2

General comments :

I find that the authors have satisfactorily addressed all my previous comments. I therefore recommend that the manuscript be accepted as is and published into ACP.